# Hippocampal astrocytes modulate anxiety-like behavior

Woo-Hyun Cho[1,7] ✉, Kyungchul Noh[1,7], Byung Hun Lee [2], Ellane Barcelon[1], Sang Beom Jun [3,4,5], Hye Yoon Park [2,6] & Sung Joong Lee[1] ✉

Astrocytes can affect animal behavior by regulating tripartite synaptic transmission, yet their influence on affective behavior remains largely unclear. Here we showed that hippocampal astrocyte calcium activity reflects mouse affective state during virtual elevated plus maze test using two-photon calcium imaging in vivo. Furthermore, optogenetic hippocampal astrocyte activation elevating intracellular calcium induced anxiolytic behaviors in astrocyte-specific channelrhodopsin 2 (ChR2) transgenic mice (hGFAP-ChR2 mice). As underlying mechanisms, we found ATP released from the activated hippocampal astrocytes increased excitatory synaptic transmission in dentate gyrus (DG) granule cells, which exerted anxiolytic effects. Our data uncover a role of hippocampal astrocytes in modulating mice anxiety-like behaviors by regulating ATP-mediated synaptic homeostasis in hippocampal DG granule cells. Thus, manipulating hippocampal astrocytes activity can be a therapeutic strategy to treat anxiety.

Astrocytes, a component of tripartite synapse, play roles in the regulation of synaptic transmission. Through their peri-synaptic astrocyte processes, they can monitor ongoing synaptic signals by neurotransmitter receptors and also affect synaptic transmission by releasing gliotransmitters. Such bi-directional communications with pre- or postsynaptic neurons were shown to regulate synaptic transmission, plasticity, and even animal behaviors[1–5].

Increasing evidence indicates a putative involvement of astrocytes in psychiatric disorder[6,7]. For instance, studies of postmortem human brain showed alteration of astrocytic morphology and volume in mood disorder[8,9]. In addition, astrocytes are implicated in anxiety-like behavior in the rat post-traumatic stress disorder (PTSD) animal model[10], and astrocytes activation in the central amygdala reduces anxiety-like behaviors in rodents[11]. It is well known that hippocampus plays a role in the development and regulation of anxiety[12]. Specifically, activation of granule cells in the dentate gyrus (DG) induces anxiolytic effects[13]. In addition, CA1 pyramidal cells in the ventral part of the hippocampus were proposed as 'anxiety-response neurons'[14]. Studies on the neural circuits proposed that hippocampus-hypothalamic or hippocampus-prefrontal circuits are involved in the anxiety-related behaviors[14,15]. Therefore, studies on anxiety have so far focused on hippocampal neurons or neural circuits involved. However, the role of hippocampal astrocytes in the manifestation of anxiety has never been formally addressed.

Thus far, the physiological role of hippocampal astrocytes has been studied mainly in memory formation, which is one of the key functions of the hippocampus. For instance, inhibition of vesicle release from astrocytes in the hippocampus reduced recognition memory formation[16], while chemogenetic hippocampal astrocytes activation enhanced contextual fear memory[4]. These studies demonstrate that hippocampal astrocytes can affect the hippocampus-dependent brain functions and behaviors. Given the control of affective behaviors by hippocampus, it is conceived that the hippocampal astrocytes may also influence behavioral responses of anxiety state.

[1]Department of Physiology and Neuroscience, Dental Research Institute, Seoul National University School of Dentistry, Seoul 08826, Republic of Korea. [2]Department of Physics and Astronomy, Seoul National University, Seoul 08826, Republic of Korea. [3]Department of Electronic and Electrical Engineering, Ewha Womans University, Seoul 03760, Republic of Korea. [4]Graduate Program in Smart Factory, Ewha Womans University, Seoul 03760, Republic of Korea. [5]Department of Brain & Cognitive Sciences, Ewha Womans University, Seoul 03760, Republic of Korea. [6]Department of Electrical and Computer Engineering, University of Minnesota, Minneapolis, MN 55455, USA. [7]These authors contributed equally: Woo-Hyun Cho, Kyungchul Noh. ✉e-mail: woohyuncho15@snu.ac.kr; sjlee87@snu.ac.kr

In this work, we explored this hypothesis and present data supporting that hippocampal astrocyte calcium activity reflect mouse anxiety state, and affects anxiety-like behavior by regulating synaptic homeostasis of DG granule cells via ATP release.

## Results

### Hippocampal astrocytes respond to anxiogenic environments

To investigate whether hippocampal astrocytes activity is associated with mouse affective state, we measured intracellular $Ca^{2+}$ activity ($\Delta F/F$) in astrocytes of the hippocampus, a brain sub-region known to regulate affective state, while mice were exploring innately anxiogenic environment. For this, we generated astrocyte-specific GCaMP6s mice (hGFAP-GCaMP6s; Fig. 1a) by crossing hGFAP-creERT2 mice expressing tamoxifen-inducible Cre recombinase under the control of GFAP promoter[17] with floxed-stop-GCaMP6s mice[18]. In these mice, GCaMP6s was expressed in various brain sub-regions including hippocampus (Supplementary Fig. 1a–c). Upon immunostaining with cell type-specific marker, GCaMP6s expression was detected only in GFAP- or S100β-positive astrocytes, but not in NeuN-positive neurons or Iba-1-positive microglia (Fig. 1b and Supplementary Fig. 1a–d).

Head-fixed awake hGFAP-GCaMP6s mice were exposed to virtual reality (VR) environment emulating either a closed or an open arm of an elevated plus maze (EPM; Fig. 1c). Mice spent much less time in the center zone in VR, which is reminiscent to the mouse behavior in open arm in EPM, thus validating the EPM test in VR (Supplementary Fig. 1e, Supplementary Data). Then, hippocampal astrocyte $Ca^{2+}$ activity was monitored in vivo by two-photon excitation microscopy (Fig. 1d). Through a cranial window made above the hippocampus, we measured $Ca^{2+}$ activity of astrocytes in stratum radiatum and stratum lacunosum-moleculare layers in the hippocampus (Fig. 1d). Astrocytes in these regions were not morphologically activated or underwent gliosis by the microsurgery for the cranial window (Supplementary fig. 1f). Interestingly, hippocampal astrocytes exhibited intracellular $Ca^{2+}$ increase while exploring the anxiogenic open center of the VR (Fig. 1e–i, Supplementary Data). The number of $Ca^{2+}$ activity peak was much higher in VR center area compared to closed or corner areas (Fig. 1f, g, Supplementary Data). Mice entering or passing through the open center exhibit significant astrocyte $Ca^{2+}$ activity, whereas mice approaching the corner did not. Specifically, 75.2% of hippocampal astrocytes showed intracellular $Ca^{2+}$ elevation when mice entered the center area and only 23.3% of hippocampal astrocytes responded when mice explored the corner area (Fig. 1h, i).

These anxiogenic environments (VR center)-responding hippocampal astrocytes were categorized into two types according to their activity patterns: pre-responsive cells ($Ca^{2+}$ increases before entering the center) and post-responsive cells ($Ca^{2+}$ increases after entering the center; Fig. 1h, i, Supplementary Fig. 2). The pre- and post-responsive cells accounted for 19.6% and 80.4% of all responsive cells, respectively (Fig. 1i). The proportions of the responsive cells were not affected by the speed of mice passing through the center area (rapidly passing (≤ 4 s) vs. slowly passing (> 4 s; Supplementary Fig. 2)). Furthermore, the number of astrocyte $Ca^{2+}$ peak in near-center zone, at which the mice first visually confront anxiogenic open center ahead (Supplementary Fig. 3a, b), is much higher than in near-corner zone in the closed area (Supplementary Fig. 3c, Supplementary Data).

To confirm that hippocampal astrocytes respond to anxiogenic open environment, not just to the change of visual images, we changed the open center image to black-walled corridor without star images. This time, there was no significant difference in the astrocyte $Ca^{2+}$ activities between center and closed areas (Supplementary Fig. 4a–c, Supplementary Data). The VR was designed to trigger anxiety by exposing the mice from dark closed environment to bright open environments. In order to test whether hippocampal astrocytes respond to bright environment under different condition, whole screen surrounding test mouse was switched from black to bright

white. As in VR center area, 67% of hippocampal cells responded to the switch to bright white screen (Supplementary Fig. 4d–f). Taken together, these results imply that hippocampal astrocytes are activated upon exposure to innately visual-based anxiogenic environment, and suggest that the $Ca^{2+}$ activity of hippocampal astrocytes discerns or reflects anxiety state.

### Optogenetic hippocampal astrocyte activation induces anxiolytic effects

Given that hippocampal astrocyte intracellular $Ca^{2+}$ activity correlates with the exposure to anxiogenic environment (Fig. 1), we explored whether hippocampal astrocyte $Ca^{2+}$ activation directly influences anxiety-like behavior. To this end, we generated astrocyte-specific channelrhodopsin 2 (ChR2) transgenic mice by crossing hGFAP-creERT2 mice with floxed-stop-ChR2-EYFP mice[19] (hGFAP-ChR2; Fig. 2a). Upon tamoxifen treatment, ChR2 expression was measured by EYFP signal. Immunostaining with two independent astrocyte markers, GFAP and S100β, showed that 74.1 ± 9.5% of hippocampal astrocytes expressed ChR2 in hGFAP-ChR2 homozygous mice (Fig. 2b, Supplementary Data). The EYFP signal was not detected in NeuN-positive neurons or in Iba-1-positive microglia in hGFAP-ChR2 mice (Supplementary fig. 5g–i), confirming that ChR2 was expressed only in astrocytes in these mice. The intensity of EYFP signal vary depending on specific brain area such as hippocampus, prefrontal cortex, striatum, amygdala, thalamus and cortex (Supplementary Fig. 5a, b). Yet, the EYFP signal was not detected in the vehicle-injected ChR2-EYFP mice or tamoxifen-injected wild-type mice (Supplementary Fig. 5c, d, Supplementary Data). The EYFP fluorescent signal in ChR2-EYFP heterozygous mice was weaker than in homozygous mice, showing a dosage effect of the *chr2* gene (Supplementary Fig. 5e, f, Supplementary Data). Based on this, homozygous mice were used for the subsequent experiments. To confirm astrocyte intracellular $Ca^{2+}$ elevation by optogenetic stimulation, primary astrocytes were cultured from hGFAP-ChR2 pups and treated with tamoxifen to induce ChR2 expression (Supplementary Fig. 5j). Blue light stimulation of these astrocytes elicited $Ca^{2+}$ signal (Fig. 2c). We also confirmed $Ca^{2+}$ elevations in adult hippocampal astrocytes upon optogenetic stimulation. We injected GFAP-jRGECO1a, an astrocyte-specific red-shifted $Ca^{2+}$ indicator, into the hippocampus of adult hGFAP-ChR2 mice. After 3 weeks, hippocampal slice was obtained and the activity of jRGECO1a-expressing astrocytes was recorded while delivering blue light stimulation (Fig. 2d, e). We observed significantly increased jRGECO1a fluorescence during optogenetic astrocyte stimulation, which returned to the basal level after turning off the light (Fig. 2f).

Since hippocampal astrocytes responded to anxiogenic stimuli by $Ca^{2+}$ activation (Fig. 1), we assessed whether optogenetic activation of hippocampal astrocytes directly alters anxiety-like behaviors in the EPM and open field test (OFT). Optic fibers were bilaterally implanted above CA1 of the dorsal hippocampus to target the stratum radiatum and stratum lacunosum-moleculare layers in the hippocampus and mice were subjected to a 15 min session of above two behavior tests (Fig. 3a). In the EPM, hGFAP-ChR2 mice exhibited a significant increase in exploration time in the open arm during a 5 min light-on epoch and spent relatively less time in the closed arms compared to control mice (Fig. 3b, c, Supplementary Fig. 6a–c, Supplementary Data). Interestingly, the increased exploration time in the open arm was maintained even after turning off the light (second 5 min light-off epoch) although slightly decreased compared to light-on epoch (Fig. 3c, Supplementary Fig. 6a, Supplementary Data). When the time spent in the open arm was measured in minute, light-induced astrocytic activation took effect 1 min after the light was turned on and was sustained for 3 min (Supplementary Fig. 6b, Supplementary Data). Likewise, the time spent in the center of EPM of hGFAP-ChR2 mice were significantly increased during light-on and second light-off epochs compared to control mice (Fig. 3d, Supplementary Data). The average speed was not different

between the two groups during the first light-off epoch, but the speed of hGFAP-ChR2 mice started to increase in the light-on epoch and further increased after turning off the light (Fig. 3e, Supplementary Data). Hippocampus is known to have heterogenic functions depending on its dorsoventral axis: dorsal hippocampus is involved in

learning and spatial memory while ventral hippocampus in affective processes[20]. To test if astrocyte's function is also dependent on the dorsoventral axis, the same experiments were performed while stimulating ventral hippocampal astrocytes. A similar pattern of behavioral change was observed in the EPM (Supplementary Fig. 6d–j,

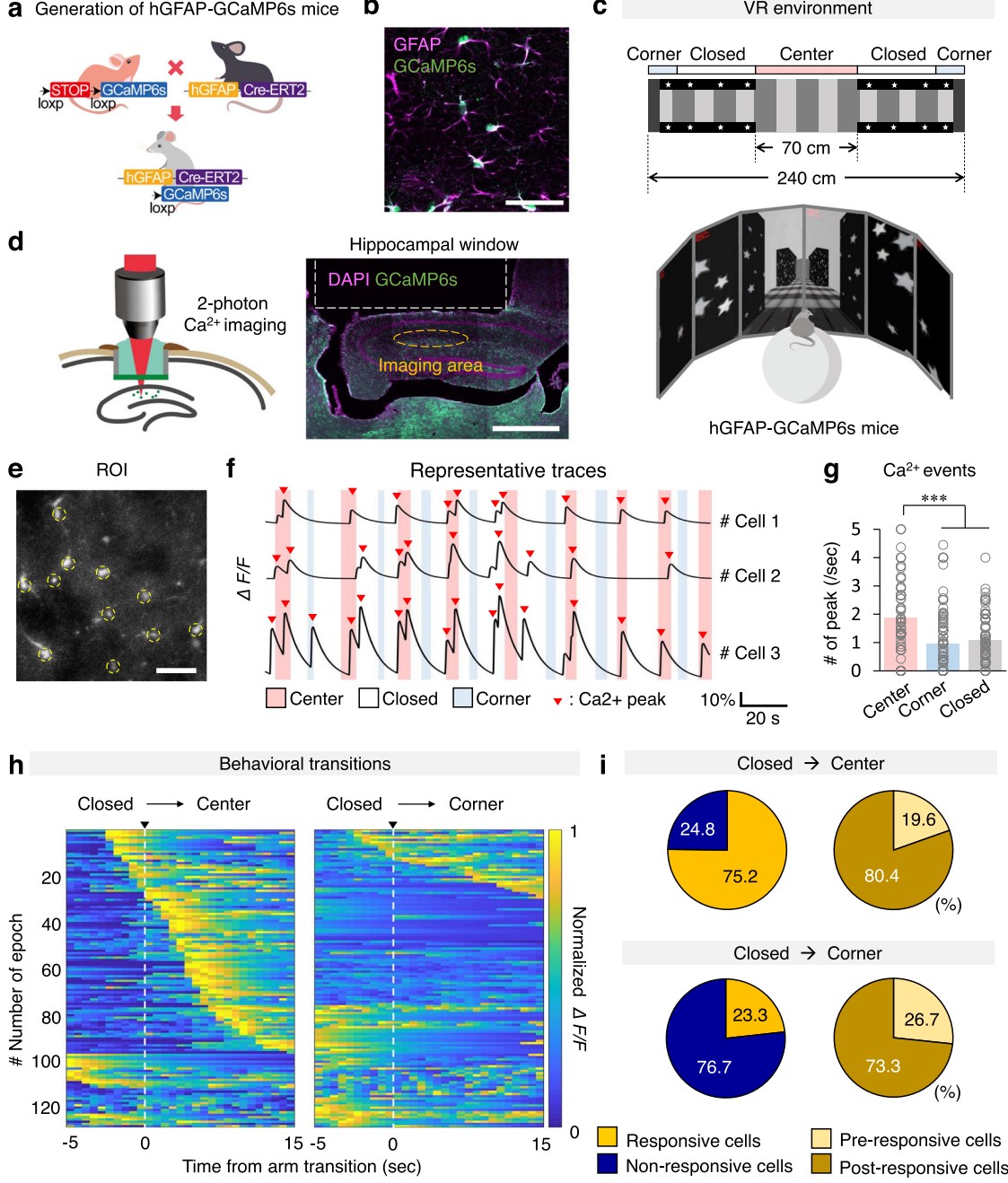

**Fig. 1 | Hippocampal astrocytes respond to anxiogenic environment by intracellular calcium elevation. a** Schematic illustration showing hGFAP-GCaMP6s generation by breeding hGFAP-CreERT2 and floxed-stop-GCaMP6s transgenic mice. **b** Representative confocal image showing GCaMP6s (green) colocalization in astrocytes (GFAP$^+$, purple). Scale bar, 50 μm. Experiment was repeated five-times independently. **c** Design of VR environment; a 2D image (upper) and 3D image (bottom) as seen by the mouse. **d** Schematic representation of two-photon imaging (left) and the hippocampal window (right) indicating the imaging area. Scale bar, 1 mm. **e** Example region of interest (ROI) of calcium signal recording. Yellow circle: recorded cells. Scale bar, 50 μm. Experiment was repeated five-times independently. **f** Representative Ca$^{2+}$ traces during exploration in VR condition. Pink, blue

and white bar indicates center, corner and closed corridor in VR. Red triangles indicate Ca$^{2+}$ peaks. **g** Number of Ca$^{2+}$ peak per second when mouse explored the VR center, corner, and closed area ($n = 87$ cells). One-way ANOVA ($p = 0.000$) followed by post-hoc analysis LSD ($p = 0.000$ (center vs corner), $p = 0.000$ (center vs closed), $p = 0.492$ (corner vs closed)). **h** Heat map trace of normalized astrocytic GCaMP6 signals when mice enter the center (left) or the corner (right) from the closed corridor ($n = 129$ events). **i** Categories and percentages of hippocampal astrocytes according to activity patterns when mice enter the center (upper) or the corner (lower) from the closed area. Black arrow: Direction of movement. The bar graphs depict the mean ± SEM. ***$p < 0.001$. See Supplementary Data for detailed statistics. Source data are provided as a Source Data file.

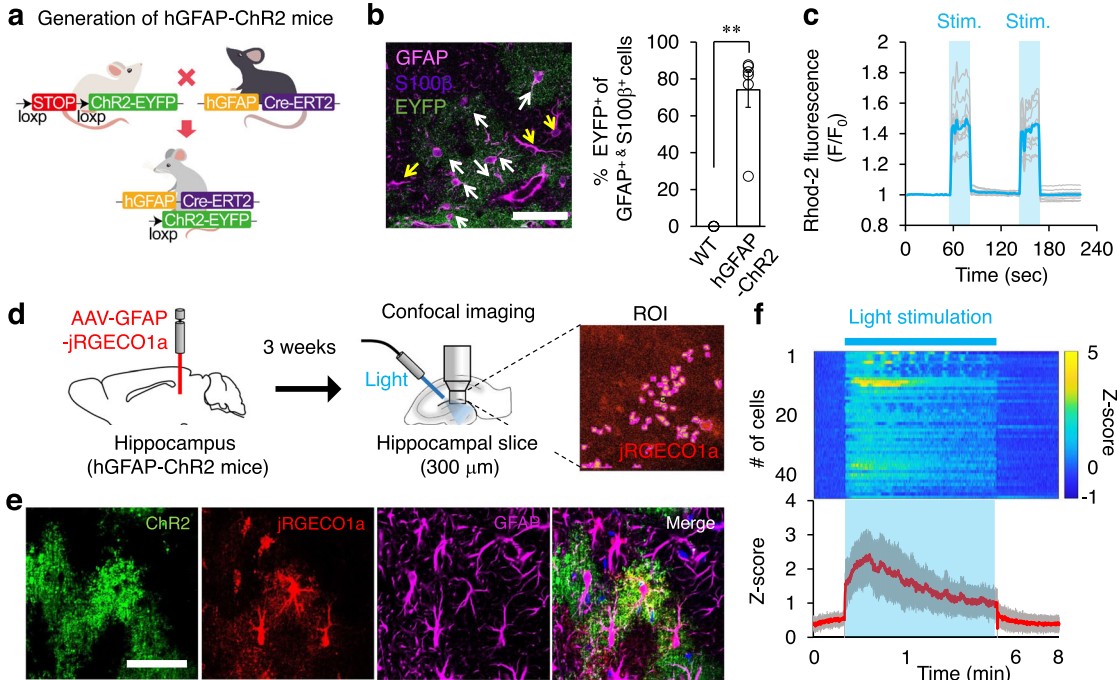

**Fig. 2 | Optogenetic astrocyte activation induces intracellular Ca²⁺ signal.**
**a** Schematic illustration showing hGFAP-ChR2 generation by breeding hGFAP-CreERT2 and floxed-stop-ChR2-EYFP transgenic mice. **b** Left, representative confocal image showing ChR2 (green: EYFP⁺) colocalization in astrocytes (pink: GFAP⁺, purple: S100ß⁺). Yellow arrow, GFAP⁺/S100ß⁺/EYFP⁻ cells ($n = 412$ cells); white arrow, GFAP⁺/S100ß⁺/EYFP⁺ cells ($n = 1206$ cells). Scale bar, 50 μm. Right, percentage of ChR2-expressing astrocytes between wild-type control ($n = 5$) and hGFAP-ChR2 ($n = 9$). Two-tailed Mann–Whitney $U$-test ($p = 0.002$). Experiment was repeated three-times independently. **c** Ca²⁺ level in ChR2-expressing primary astrocytes

with or without light stimulation (blue: mean trace, gray: individual activities).
**d** Schematic illustration of experiment for astrocytic Ca²⁺ imaging in acute hippocampal slice. Right image, ROI for jRGECO1a-positive astrocytes. **e** Expression of ChR2 (green), jRGECO1a (red), and GFAP (pink) in hippocampus. Scale bar, 50 μm.
**f** Heatmap representation and averaged trace of hippocampal astrocyte Ca²⁺ activities while delivering continuous light stimulation for 5 min ($n = 46$ cells, red: average, gray: standard error). Representative data from three independent experiments. **$p < 0.01$. See Supplementary Data for detailed statistics. Source data are provided as a Source Data file.

---

Supplementary Data) showing that optogenetic hippocampal astrocyte activation modulates anxiety-like behaviors in the EPM regardless of the heterogenic nature of the hippocampal dorsoventral axis.

In the OFT, light-induced astrocyte activation in the dorsal hippocampus led to a dramatic increase in exploration based on total distance traveled compared with control mice or distance during the first light-off epoch (Fig. 3f, g, Supplementary Fig. 7a, Supplementary Data). This increased exploration of hGFAP-ChR2 mice continued after turning off the light (Fig. 3g, Supplementary Fig. 7a). hGFAP-ChR2 mice showed a significantly larger difference in exploration distance between the first light-off epoch and the light-on epoch and between the last light-off epoch and the first light-off epoch compared with control mice (Fig. 3h, Supplementary Data). However, significant change was not observed in the distance traveled in the center area between the light-on and light-off epochs (Supplementary Fig. 7b, Supplementary Data). In addition to greater exploration activity, mice showed elevated speed during the 5 min light-on epoch that further increased during the second 5 min light-off epoch (Supplementary Fig. 7c, Supplementary Data). To check how long the light-stimulated exploration activity increase is sustained, mice were exposed to open field apparatus for 1 h. The duration of astrocyte activation-induced exploration drive was maintained for 10 min after the light was off and gradually decreased to the control level (Fig. 3i, Supplementary Data). Optogenetic astrocyte activation in the ventral hippocampus showed similar behavioral changes in the OFT (Supplementary Fig. 7d–i, Supplementary Data). Taken together, these data suggest that optogenetic hippocampal astrocyte activation modulates anxiety-like behaviors in EPM and OFT regardless of the heterogeneous nature of the hippocampus through the dorsoventral axis.

Hippocampus is a key brain region for spatial memory formation. Therefore, we tested whether optogenetic hippocampal astrocyte activation affects these hippocampus-dependent brain functions in T-maze test. Hippocampal astrocytes received 5 min of light stimulation at the beginning of a 20 min exposure time in the T-maze. Optogenetic astrocyte activation did not have any effect on the ability to distinguish the old and new arms: both hGFAP-ChR2 and control mice could distinguish the new arm (Supplementary Fig. 8a–c, Supplementary Data). Next, we tested whether astrocyte stimulation induces place preference in the real-time place preference test. hGFAP-ChR2 mice did not have preference for either chamber (Supplementary Fig. 8d, e, Supplementary Data). Our data show that optogenetic astrocyte activation does not affect hippocampus-dependent spatial working memory formation or place preference, indicating that hippocampal astrocyte activity specifically modulates anxiety-related hippocampal function.

## Optogenetic astrocyte stimulation enhances hippocampal neuronal activity by increasing sEPSC frequency
To assess whether optogenetic astrocyte activation affects neuronal activity[1,3], an optical fiber was implanted in the dorsal hippocampus and illuminated with blue light for 5 min (Fig. 4a). After 1 h, neuronal activation was monitored using c-Fos immunostaining in hippocampus. The blue light stimulation dramatically increased c-Fos level in ipsilateral DG of hGFAP-ChR2 mice (Fig. 4a, b, Supplementary Data) and slightly increased c-Fos level in the CA1 and CA3 (Supplementary Fig. 9a). Notably, optogenetic astrocyte activation of the dorsal hippocampus induced c-Fos expression in ventral hippocampal neurons, and vice versa (Supplementary Fig. 9b, Supplementary Data). To exclude a possibility of long-range light projection effect in hGFAP-ChR2

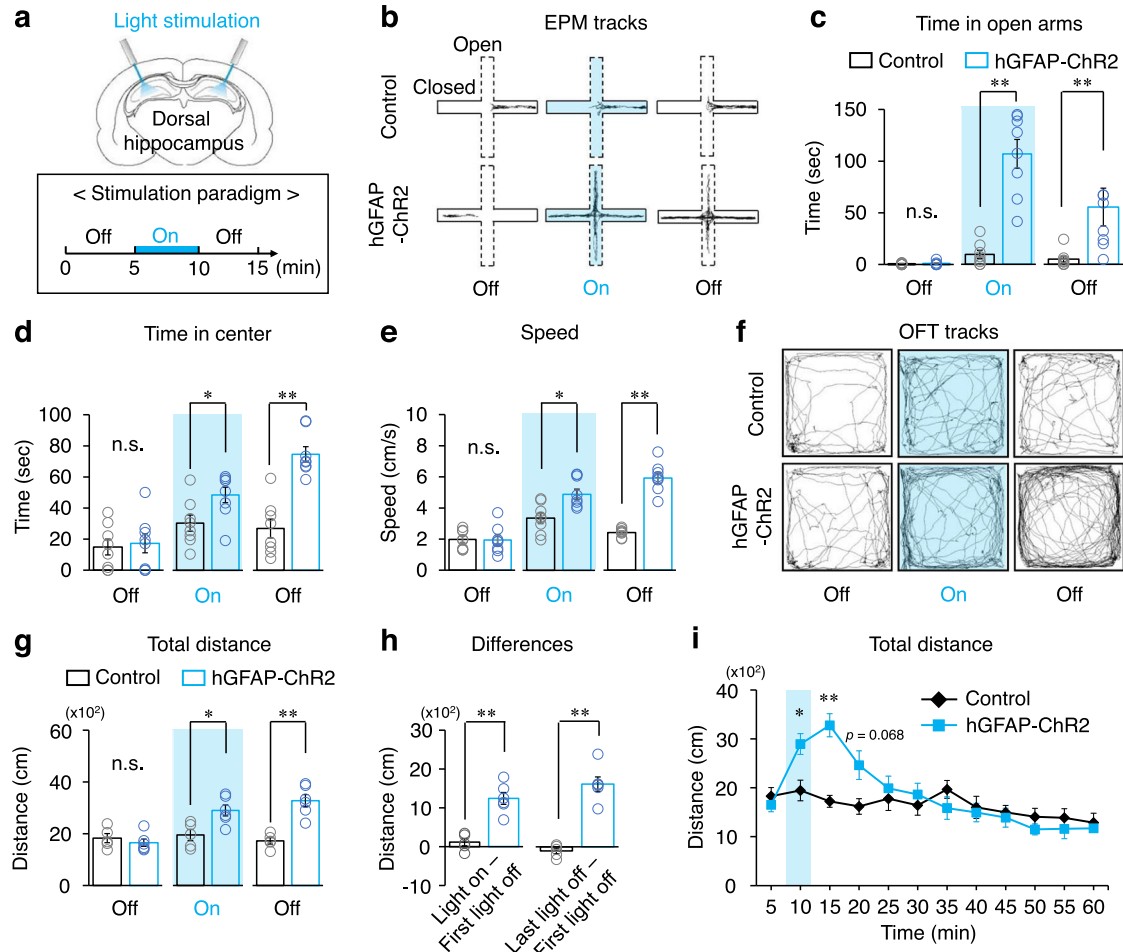

**Fig. 3 | Optogenetic astrocyte activation induces anxiolytic effects and increases exploratory drive. a–e** EPM test with light stimulation of dorsal hippocampal astrocytes between control and hGFAP-ChR2 ($n = 8$ mice in each group). Time in open arm/center, and speed are analyzed from this experiment. **a** Schematic illustration of optical fiber implantation in dorsal hippocampus (upper) and light stimulation (lower). **b** Representative traces of control and hGFAP-ChR2 mice in EPM. Blue color represents light-on epoch. **c** Time spent in the open arm by control and hGFAP-ChR2 mice. Two-way repeated ANOVA ($p = 0.000$), followed by two-tailed Mann−Whitney $U$-test ($p = 0.817$ (first light-off), $p = 0.001$ (light-on), $p = 0.003$ (last light-off)). **d**,Time spent in the center by control and hGFAP-ChR2, two-way repeated ANOVA ($p = 0.002$), followed by two-tailed Mann−Whitney $U$-test ($p = 0.832$ (first light-off), $p = 0.046$ (light-on), $p = 0.001$ (last light-off)). **e** Travel speed by control and hGFAP-ChR2. Two-way repeated ANOVA ($p = 0.002$), followed by two-tailed Mann−Whitney $U$-test ($p = 0.674$ (first light-off),

$p = 0.010$ (light-on), $p = 0.001$ (last light-off)). **f–i** OFT test with light stimulation of dorsal hippocampal astrocytes between control ($n = 5$ mice) and hGFAP-ChR2 ($n = 6$ mice). Total distance and differences are analyzed from this experiment. **f** Representative traces of control and hGFAP-ChR2 mice during OFT. **g** Total distance traveled by control and hGFAP-ChR2. Two-way repeated ANOVA ($p = 0.007$), followed by two-tailed Mann−Whitney $U$-test ($p = 0.361$ (first light-off), $p = 0.018$ (light-on), $p = 0.006$ (last light-off)). **h** Difference of distance traveled in OFT of control and hGFAP-ChR2. Left, during light-on and first light-off epoch. Right, during last light-off and first light-off epoch. Two-tailed Mann−Whitney $U$-test ($p=0.006$). **i** Total distance traveled by control and hGFAP-ChR2 mice upon 5-min light stimulation measured for 1 h. Two-tailed Mann−Whitney $U$-test ($p = 0.018$ (5–10 min), $p = 0.006$ (10–15 min)). The bar graphs depict data as mean ± SEM. $*p < 0.05$; $**p < 0.01$. n.s., not significant. See Supplementary Data for detailed statistics. Source data are provided as a Source Data file.

mice, we injected AAV-GFAP-ChR2-EYPF virus locally in dorsal hippocampus and stimulated blue light (Supplementary Fig. 9c). As in the hGFAP-ChR2 mice, optogenetic astrocyte activation of dorsal hippocampus resulted in c-Fos activation in both dorsal and ventral hippocampus (Supplementary Fig. 9c). These data indicate that local optogenetic astrocyte stimulation affects neurons across a wide hippocampal area.

To test the functional consequences of astrocyte activation on neuronal activity, whole-cell recording was performed on DG granule cells in acute hippocampal slice of hGFAP-ChR2 mice. Optogenetic astrocyte activation with blue light stimulation increased the frequency, but not the amplitude, of spontaneous excitatory postsynaptic currents (sEPSCs; Fig. 4c–f, Supplementary Data). The sEPSC frequency instantaneously increased, peaked within the first 20–50 s, and gradually returned to basal level during the last 4 min of light stimulation (Fig. 4f). The effects of

optogenetic astrocyte activation on synaptic transmission of DG granule cells was independent of hippocampal dorsoventral axis: optogenetic astrocyte stimulation increased sEPSC frequency in both dorsal and ventral hippocampus (Supplementary Fig. 10a–d, Supplementary Data). Similarly, optogenetic astrocyte activation increased sEPSC frequency of CA1 pyramidal neurons (Supplementary Fig. 10e, f, Supplementary Data). To rule out the possibility of neuronal damage by 5-min continuous light stimulation, optogenetic astrocyte activation was applied to dorsal and ventral hippocampal slice two times while recording DG granule cells. The second light stimulation increased the frequency of sEPSC comparable levels as the first light stimulation in both dorsal and ventral hippocampus (Supplementary Fig. 10g, h, Supplementary Data). Taken together, these data show that optogenetic astrocyte activation enhances hippocampal neuronal activity by increasing sEPSC frequency without affecting neuronal damage.

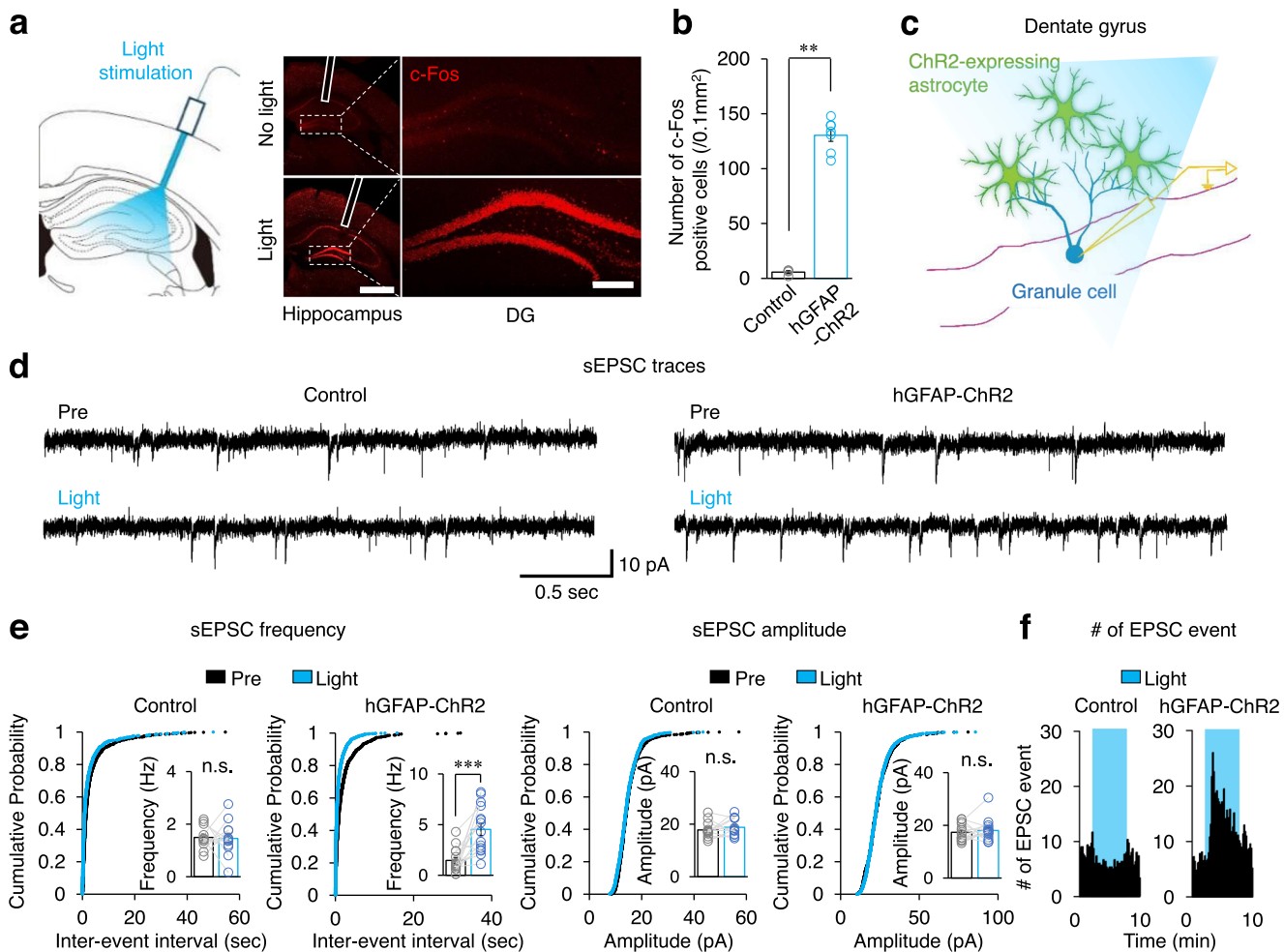

**Fig. 4 | Optogenetic astrocyte stimulation enhances hippocampal neuronal activity by increasing sEPSC frequency. a** Left, schematic illustration of optical fiber implantation and light stimulation in the hippocampus. Right, light stimulation increased c-Fos (red) expression in the hippocampus of hGFAP-ChR2 mice. White lines, track of implantation. Scale bars, 1 mm (left), 200 μm (right). **b** Quantification of the number of c-Fos positive cells in DG between control ($n = 4$) and hGFAP-ChR2 ($n = 7$). Two-tailed Mann–Whitney $U$-test ($p = 0.008$). **c** Schematic illustration of whole-cell voltage clamp recording of DG granule cells while optically stimulating nearby astrocytes. **d** Representative sEPSC traces measured from control and hGFAP-ChR2 hippocampus before (Pre) and during (Light) stimulation. Scale bar, 0.5 s and 10 pA. **e** sEPSC frequency and amplitude in control ($n = 12$ cells) and hGFAP-ChR2 ($n = 15$ cells). sEPSC frequency was compared using Paired $t$-test ($p = 0.879$ (Pre vs Light in control group) and $p = 0.000$ (Pre vs Light in hGFAP-ChR2)). Cumulative probabilities were measured by two sample Kolmogorov-Smirnov test ($p = 0.813$ (pre vs light in control group), and $p = 0.000$ (pre vs light in hGFAP-ChR2)). sEPSC amplitude was compared using Paired $t$-test. ($p = 0.436$ (pre vs light in control group), and $p = 0.566$ (Pre vs Light in hGFAP-ChR2)). Cumulative probabilities were measured by two sample Kolmogorov–Smirnov test ($p = 0.813$ (Pre vs Light in control group), and $p = 0.171$ (Pre vs Light in hGFAP-ChR2)). **f** Number of sEPSC events was increased by light stimulation. Blue box, period of light stimulation (5 min). The bar graphs depict data as mean ± SEM. **$p < 0.01$; ***$p < 0.001$. n.s. not significant. See Supplementary Data for detailed statistics. Source data are provided as a Source Data file.

A previous study suggested that optogenetic astrocyte stimulation may affect neuronal activity by increasing extracellular $K^+$ concentration ($[K^+]_o$) of medium spiny neurons in striatum[21]. To test similar mechanism is involved in hippocampus, we measured neuronal excitability of DG granule cells while stimulating hippocampal astrocyte. Although the threshold current to trigger action potential was slightly reduced by astrocytes stimulation (Supplementary Fig. 11a, b, Supplementary Data), the excitability during the last second was not different between control and hGFAP-ChR2 mice (Supplementary Fig. 11c, Supplementary Data) indicating that firing rate was not changed by optogenetic astrocyte activation. Meanwhile, during recording sEPSC, we observed significantly increased inward current (Supplementary Fig. 11d), which might be due to the elevated $[K^+]_o$ by optogenetic astrocyte activation[21]. To test if the optogenetic astrocyte activation-induced $[K^+]_o$ increase might have led the synaptic activity changes observed in our study, we recorded sEPSC of DG granule cells while increasing $[K^+]_o$ by 2 and 4 mM in extracellular ACSF buffer,

which are comparable to the $[K^+]_o$ increase by optogenetic astrocyte activation[21]. While $[K^+]_o$ increase by 2 mM (7 mM final concentration) elevated inward currents, it did not affect the frequency or amplitude of sEPSC (Supplementary Fig. 11e–g, Supplementary Data). Although a slight sEPSC frequency increase was observed at 9 mM of $[K^+]_o$, this time amplitude of sEPSC was also increased (Supplementary Fig. 11h–j, Supplementary Data). These results indicate that optogenetic astrocyte activation-induced $[K^+]_o$ increase does not fully explain the synaptic change observed in our study, thus implicate other mechanism such as synaptic regulation by gliotransmitter in the optogenetic astrocyte activation-induced mouse anxiolytic effects.

**Optogenetic astrocyte activation induces anxiolytic behavior by ATP release**

Therefore, we searched for putative gliotransmitters released upon optogenetic astrocyte activation[22–24]. Light stimulation released ATP from the ChR2-expressing primary astrocytes cultured from whole

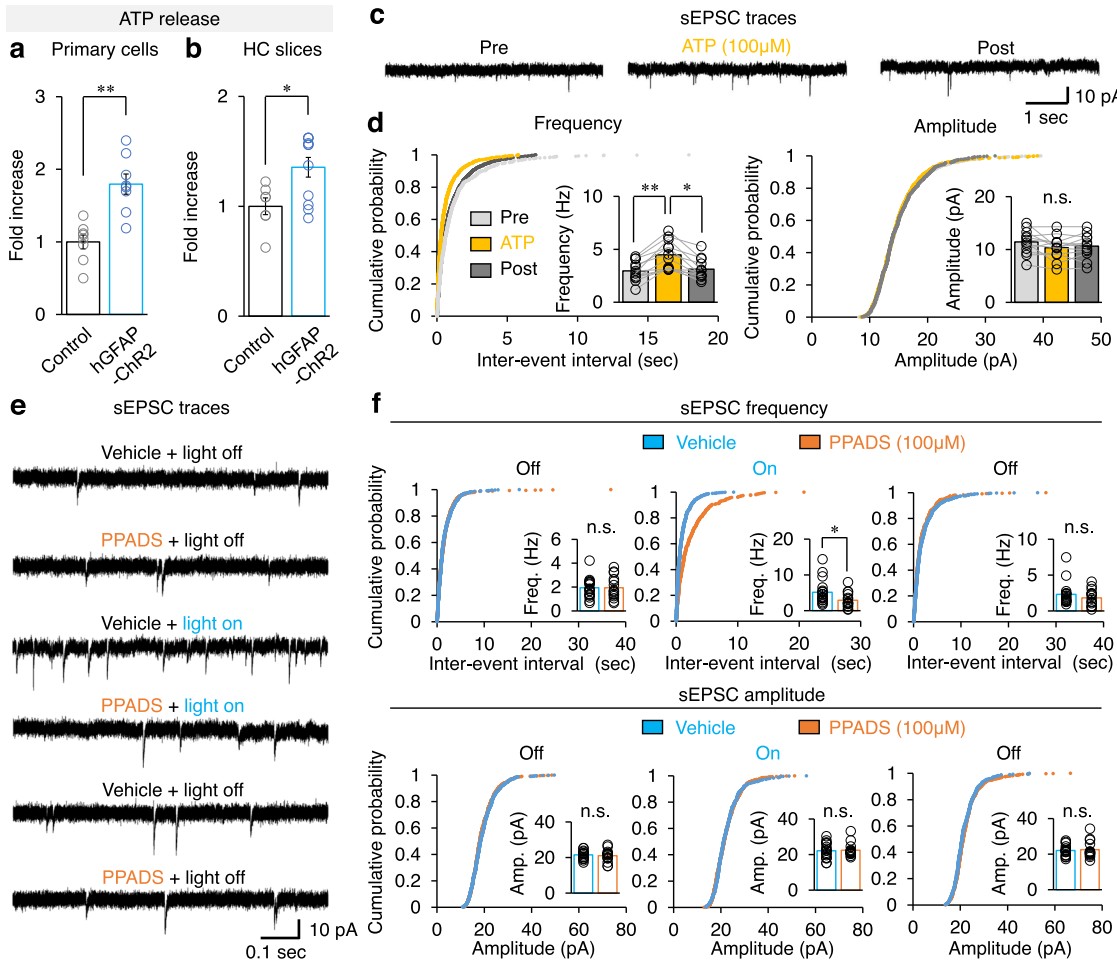

**Fig. 5 | ATP released from activated astrocyte increases hippocampal sEPSC.**
**a** ATP release from primary astrocyte cells of wild-type control and hGFAP-ChR2 mice ($n = 8$ mice in each group). Two-tailed Mann-Whitney $U$-test, $p = 0.001$. **b** ATP release from hippocampal slices of control ($n = 7$ mice) and hGFAP-ChR2 ($n = 11$ mice). Two-tailed Mann-Whitney $U$-test, $p = 0.044$. HC, hippocampus.
**c** Representative sEPSC traces measured from hippocampal DG granule cells before (Pre), 100 μM of ATP, and after (Post) treatment. Scale bar, 1 s and 10 pA. **d** sEPSC frequency (left) and amplitude (right) of hippocampal DG granule cells ($n = 12$ cells) upon ATP treatment. One-way ANOVA for sEPSC frequency ($p = 0.004$), followed by Bonferroni post hoc analysis ($p = 0.007$ (Pre vs ATP), $p = 1.000$ (Pre vs Post), $p = 0.017$ (ATP vs Post)). One-way ANOVA for sEPSC amplitude ($p = 0.485$).

**e** Representative sEPSC traces measured from hGFAP-ChR2 hippocampus during treatment of vehicle or PPADS before (light-off), during (light-on), and after (light-off) light stimulation. Scale bar, 0.1 s and 10 pA. **f** sEPSC frequency and amplitude of DG granule cells between vehicle group ($n = 18$ cells) and PPADS ($n = 15$ cells). sEPSC frequencies were compared using two-tailed Mann-Whitney $U$-test ($p = 0.828$ (first light-off), $p = 0.021$ (light-on), $p = 0.347$ (last light-off)). sEPSC amplitudes were compared using two-tailed Mann-Whitney $U$-test ($p = 0.691$ (first light-off), $p = 0.942$ (light-on), $p = 0.664$ (last light-off)). *$p < 0.05$; **$p < 0.01$. n.s., not significant. See Supplementary Data for detailed statistics. Source data are provided as a Source Data file.

cerebrum (Fig. 5a, Supplementary Data), hippocampus and cortex (Supplementary Fig. 12a). Likewise, optogenetic astrocyte activation induced ATP release in hippocampal slice (Fig. 5b, Supplementary Data). However, glutamate or D-serine were not released in hippocampal slice (Supplementary Fig. 12b, c, Supplementary Data). These data are in agreement with a previous report showing ATP release due to optogenetic astrocyte stimulation[22]. To test whether ATP renders the same synaptic activity change as optogenetic astrocyte stimulation, we applied ATP in extracellular ACSF buffer and recorded sEPSC in hippocampal DG granule cells. The sEPSC frequency, not amplitude, increased by ATP treatments (Fig. 5c, d, Supplementary Data). Additionally, pyridoxalphosphate-6-azophenyl-2',4'-disulfonic acid (PPADS), a purinergic receptor antagonist, abolished light-induced sEPSC frequency increase (Fig. 5e, f, Supplementary Data). These data indicate that ATP receptor signaling is responsible for optogenetic astrocyte activation-driven sEPSC frequency increase.

To test whether astrocyte-released ATP is responsible for the behavioral changes observed upon optogenetic astrocyte activation, we delivered PPADS and light stimulation simultaneously in mice

during EPM test (Fig. 6a). Mice behaviors were tested in the EPM with or without PPADS administration at a 48-h interval (Pre, PPADS, Post). Before PPADS injection, optogenetic astrocyte activation robustly increased the time spent in the open arm in the EPM, which was almost completely abolished upon PPADS administration (Fig. 6b, c, Supplementary Data). However, the increase in travel speed or time spent in the center due to optogenetic astrocyte activation was not affected by PPADS administration (Fig. 6d, e, Supplementary Data). These data indicate that ATP signaling is required for astrocyte activation-induced explorative drive in the open arm, but not for the increased speed and time spent in the center of the EPM. Notably, PPADS injection significantly attenuated the maintenance of the light-induced increase in time spent in the center or travel speed during the second light-off epoch (Fig. 6d, e, Supplementary Data). These data indicate that ATP signaling is required for maintenance but not induction of the increase in speed and time in the center after optogenetic astrocyte activation. However, optogenetic astrocyte activation-induced increase of total travel distance or speed in the OFT was not affected by PPDAS injection (Supplementary Fig. 12e−i, Supplementary Data). Taken together, these

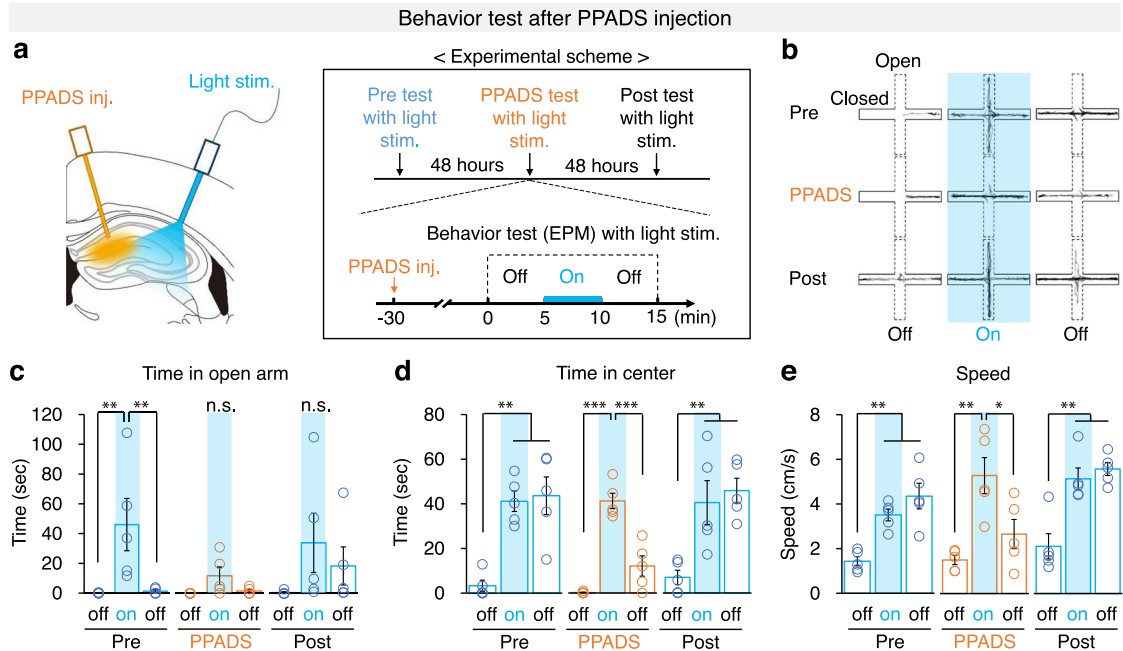

**Fig. 6 | Optogenetic astrocyte activation induces anxiolytic behavior by ATP release. a** Illustration of PPADS administration and optical stimulation in dorsal hippocampus (left) and the experimental scheme (right). **b**–**e** EPM test with light stimulation of hippocampal astrocytes while PPADS treatment between control and hGFAP-ChR2 (n = 5 mice in each group). Time in open arm/center, and speed are analyzed from this experiment. **b** Representative traces in the EPM. **c** Time spent in open arms during Pre, PPADS, and Post treatment and light stimulations. One-way ANOVA for Pre treatment (p = 0.012) followed by post hoc analysis LSD (p = 0.008 (first light-off vs light-on), p = 0.009 (light-on vs last light-off)); One-way ANOVA for light epochs in PPADS treatment, and post-treatment. **d**, Time in the center area during pre, PPADS and post treatments and light stimulations. One-way ANOVA for Pre treatment (p = 0.000) followed by LSD (p = 0.001 (first light-off vs light-on), p = 0.000 (first light-off vs last light-off)); One-way ANOVA for PPADS treatment

(p = 0.000) followed by LSD (p = 0.000 (first light-off vs light-on), p = 0.026 (first light-off vs last light-off), p = 0.000 (light-on vs last light-off)); One-way ANOVA for Post treatment (p = 0.003) followed by LSD (p = 0.004 (first light-off vs light-on), p = 0.002 (first light-off vs last light-off)). **e** Speed of mice during Pre, PPADS and Post treatments and light stimulations. One-way ANOVA for Pre treatment (p = 0.001) followed by LSD (p = 0.003 (first light-off vs light-on), p = 0.000 (first light-off vs last light-off)); One-way ANOVA for PPADS treatment (p = 0.003) followed by LSD (p = 0.001 (first light-off vs light-on), p = 0.010 (light-on vs last light-off)); One-way ANOVA for Post treatment (p = 0.000) followed by LSD (p=0.001 (first light-off vs light-on), p = 0.000 (first light-off vs last light-off)). The bar graphs depict the mean ± SEM. *p < 0.05; **p < 0.01; ***p < 0.001. n.s., not significant. See Supplementary Data for detailed statistics. Source data are provided as a Source Data file.

data indicate that optogenetic hippocampal astrocyte activation leads to release of ATP, which in turn modulates hippocampal neuron activity and subsequently anxiety-like behavior (Supplementary Fig. 13).

## Discussion

In this study, we show that hippocampal astrocytes can respond to anxiogenic environment and, furthermore, directly modulate mice anxiety-related behaviors in vivo. Abnormal and uncontrolled anxious state is a key affective feature of PTSD or major depressive disorder (MDD), and astrocytes have long been implicated in these disorders. For instance, decreased astrocyte cell number and density was detected in hippocampus or prefrontal cortex of postmortem brain of MDD patients[8,9]. In line with this, in an animal PTSD model, rats manifesting anxiety-like behavior after prolonged stress had reduced level of GFAP in the hippocampus[10]. In addition, anti-psychiatric drugs such as riluzole or fluoxetine can rescue the stress-induced reduction of astrocyte cell number[25,26]. These prior studies suggested that astrocyte may be involved in the development of affective and mood disorders[8,9,24–26]. However, the putative mechanisms how brain astrocytes influence affective state have not been resolved. In this regard, our study proposes a mechanism of affective brain functions such as anxiety regulated by hippocampal astrocyte activity.

By using two-photon calcium imaging in awake head-fixed mice in VR, we discovered that hippocampal astrocytes respond to anxiogenic environment by elevating intracellular calcium signals. Although the outcome of the hippocampal astrocyte calcium signals in anxiogenic environment is not fully elucidated, considering optogenetic astrocyte activation eliciting intracellular calcium signals reduce anxiety-like

behaviors, hippocampal astrocyte activation may contribute to overcome or reduce innate anxiety. In this regard, our data offer a perspective on why chronic stress-exposed mice, of which astrocyte cell number is reduced in hippocampus[25], could not overcome innate anxiety and manifest anxiety-like behaviors. In this study, it was not elucidated how hippocampal astrocytes respond to anxiogenic stimuli. Astrocytes express diverse Gq-coupled metabotropic receptors for neurotransmitters or neuropeptides, which can trigger intracellular $Ca^{2+}$ elevation. Therefore, we can speculate that certain neurotransmitter/neuropeptide released in the hippocampus upon anxiogenic stimuli may lead to astrocyte $Ca^{2+}$ elevation in physiological condition. In this regard, it is of note that hippocampal astrocytes express oxytocin receptor[27], and oxytocin released in amygdala activate astrocyte $Ca^{2+}$ signal through oxytocin receptor and mediate its anxiolytic effects[11]. Besides, astrocytes express receptors for dopamine and serotonin, key neuropeptides regulating mood and emotion[28,29]. Therefore, it can be speculated that these neuropeptides exert their effects by regulating hippocampal astrocyte activity, in addition to their direct effects on neuronal dopamine/serotonin receptors.

In our study, we introduced anxiogenic environment to mice via VR; the corner and closed areas of VR were designed to simulate dark walls of closed arm in EPM, while the center area was open and bright giving a sense of uneasiness. VR is widely utilized to investigate hippocampal functions, which allows in vivo imaging using two-photon microscopy and simultaneous spatial navigation behaviors[30,31]. Although there are limitations, the tendency of mice to spend more time in closed or corner area than open center area in VR indicates that mice on VR open center feels anxiety as they were in real EPM.

We identified ATP as a gliotransmitter responsible for the changes in hippocampal synaptic activity and the anxiolytic behavior. The increases of sEPSC frequency and neuronal excitability by ATP have been reported in trigeminal mesencephalic nucleus[32] and hippocampus[33], respectively. In our study, pharmacological inhibition by PPADS completely abrogated the optogenetic astrocyte activation-induced hippocampal synaptic activity changes. However, in mouse behavior, it inhibited only the duration of stay in open arms in EPM, but not the time spent in center area or speed (Fig. 6). Also, in the OFT, PPADS did not block the increase of exploratory drive and speed (Supplementary Fig. 12). This might be simply due to difference in local concentration of PPADS in vivo vs. in vitro. We can also conjecture that ATP and purinergic receptor signaling is involved in facing potential threats but other yet unidentified gliotransmitter or PPADS-insensitive adenosine receptors are involved in the increase in speed or exploratory drive observed in the astrocyte-stimulated mice. Interestingly, the anxiolytic behavioral changes due to optogenetic astrocyte activation was maintained even after turning off the light, which was also blocked by PPADS treatment. There is no clear explanation for these long-lasting anxiolytic effects. Although there is a limitation of cellular resolution in our ATP experiments, it is conceivable that hippocampal astrocyte-derived ATP and its downstream purinergic receptor signaling may render long-lasting behavioral effects by influencing DG synaptic plasticity of mossy fiber-CA3 synapse, which warrants future investigation.

Activated astrocytes release various gliotransmitters, including glutamate[34] and D-serine[35]. Although we did not detect a significant increase in glutamate or D-serine in our study, we cannot exclude the possibility that these gliotransmitters are released from hippocampal astrocytes in response to other stimuli or behavioral contexts and thereby affect other hippocampus-dependent behaviors[35,36]. Indeed, previous studies have revealed a role of hippocampal astrocytes in memory functions. For instance, the inhibition of vesicle release from astrocytes in the hippocampus reduces the duration of gamma oscillation and recognition memory[16]. In addition, astrocytic Gq pathway activation enhances contextual fear conditioning[4], and astrocytic Gi pathway activation during the memory acquisition phase impairs remote memory recall by disrupting CA3 to CA1 communication[37]. Although optogenetic hippocampal astrocyte activation did not impair T-maze behavior (Supplementary Fig. 8a–c) in our study, it may have affected memory performance accompanying anxiety. Moreover, hippocampal neurons project to distal brain regions, such as the prefrontal cortex, and regulate mouse social behaviors[38]. Considering that optogenetic astrocyte stimulation enhanced excitatory synaptic transmission in the hippocampus in our study, it is possible that hippocampal astrocytes modulate neuronal outputs to the prefrontal cortex during social behavior and thereby affect mouse social behaviors, which warrants future investigation.

It is generally considered that hippocampal neurons play distinct role depending on the dorsoventral axis; neurons of dorsal pole hippocampus contribute spatial and contextual memory[20] whereas neurons of ventral pole are involved in anxiety-related behavior[14]. However, in our study, we found that hippocampal astrocytes, not only of ventral but also of dorsal, modulates affective brain function. Our data are in line with a study by Kheirbek et al.[13] who showed that optogenetic stimulation of DG granule cells in both dorsal and ventral exerts anxiolytic effects. Given this, it is conceived that hippocampal astrocyte activation at dorsal and ventral hippocampus render anxiolytic effects by elevating synaptic activity of dorsal and ventral DG granule cells, respectively. Interestingly, in our study, local optogenetic astrocyte stimulation affected neurons across a wide hippocampal area, dorsal and ventral. Although speculative, we can conjecture that groups of astrocytes may regulate distant neuronal activity via gap-junction connectivity[39], therefore information received by dorsal hippocampal astrocyte can be relayed to ventral hippocampal astrocyte and vice versa. Alternatively, DG granule cells receiving signals from astrocyte in molecule layer might send out recurrent excitatory signal to both dorsal and ventral DG granule cells, thus granule cell activation in dorsal hippocampus is able to activate cells in ventral hippocampus, and vice versa.

Conclusively, we reveal an unexpected role of hippocampal astrocytes that respond to anxiogenic environment and modulate mice anxiety-like behavior by regulating synaptic activity of granule cells via ATP release. These data may offer a strategy to treat anxiety disorders by targeting hippocampal astrocytes.

## Methods

### Animals

All animal studies were approved and guided by the Seoul National University Institutional Animal Care and Use Committee (SNU IACUC; protocol #, SNU-171115-6-3). We used 8 to 16-week-old C57BL/6 J male mice (DooYoel Biotech Co., Seoul, Korea). Floxed-stop-GCaMP6s and Floxed-stop-ChR2(H134R)-EYFP mice were purchased from The Jackson Laboratory (Stock no. 028866 & 12569, Bar Harbor, ME, USA). hGFAP-CreERT2 mice were obtained from the laboratory of Dr. Frank Kirchhoff (Max Plank Institute, Munich, Germany). For optogenetic modulation of astrocytic activity, floxed-stop-GCaMP6s and floxed-stop-ChR2(H134R)-EYFP mice were crossed with hGFAP-CreERT2 mice (hGFAP-GCaMP6s and hGFAP-ChR2). All male hGFAP-CreERT2 mice were administered tamoxifen (i.p. 100 µg/g) for 8 consecutive days (i.p. 100 µg/g) starting at 8 weeks old. For the control group, vehicle (90% corn oil and 10% ethanol) was administered at the same volume as tamoxifen. All experiments were conducted within 6 weeks of the first tamoxifen injection. The animals were housed and maintained in a controlled environment at 22 °C–24 °C and 55% humidity with 12 h light/dark cycles and fed regular rodent chow and tap water *ad libitum*.

### Two-photon $Ca^{2+}$ imaging

**Cranial window surgery.** For in vivo imaging, we used 10-week-old hGFAP-GCaMP6s mice. For cranial window surgery[40], the mice were first anesthetized by intraperitoneal injection of ketamine/xylazine cocktail (100 µg/g, 10 µg/g). Using a trephine drill (FST, 1800427, Foster City, CA, USA), an -2.7-diameter craniotomy was created at AP −2.0 mm, ML + 1.8 mm from the bregma. The cortical tissue overlying the hippocampus was carefully aspirated and a coverslip attached to a cylindrical stainless-steel cannula (diameter, 2.7 mm; height, 1.3 mm) was inserted into the craniotomy. The space between the cannula and craniotomy was sealed with tissue adhesive (3 M, Vetbond, St. Paul, MN, USA). A customized head ring was then attached to the dried skull using a self-curing adhesive resin cement (Sun Medical, Super Bond C&B, Shiga, Japan).

**Virtual reality experiment (VR) and two-photon imaging.** Three days after the cranial window surgery, the mice were put on a water restriction schedule (1 ml/day) for 5–7 days to subsequently administer water as a reward. JetBall thin-film transistor (TFT) system (PhenoSys, Berlin, Germany) was used for VR experiment, which consisted of a six-panel monitor, a spherical treadmill, and a water reward system. We inserted a needle into the side of the treadmill to restrict its movement into one dimension. The VR training was then performed on an infinite linear track that rewards water at random locations. The training lasted 0.5–1 h per day for -14 days, and was considered completed when the mice successfully traveled more than 50 m in 30 min. The main experiment was then conducted on the following day. The virtual environment for the main experiment was 2.4 m long and 0.3 m wide, and consisted of three-side black walls at both ends. Once the mice reached the end of the environment (±1 m), the needle was released from the treadmill, allowing the mice to rotate in the VR. When the mice turned 180° at the end, the needle was inserted into the treadmill again.

For in vivo imaging, we used a two-photon excitation laser scanning microscope (Olympus, FVMPE-RS, Tokyo, Japan), which consisted of two GaAsP photomultiplier tubes, a Ti/Sapphire laser (Mai-Tai® DeepSee, Spectra-Physics, Andover, MA, USA), a galvo/resonant scanner, and a 25 × 0.95 numerical aperture water immersion objective with an 8 mm working distance (Olympus, XLSLPLN25XSVMP2). We targeted cells at a depth of 50-200 μm beneath the CA1 pyramidal layer. To image GCaMP6s signal of astrocytes, we scanned an area of 341 × 341 μm at 30 Hz using a laser tuned to 900 nm wavelength and a resonant scanner. The two-photon imaging was started by a triggering signal from the VR system that allowed simultaneous recording of the VR position and calcium activity.

**Image registration and calcium signal extraction.** To correct the motion artifact and enhance the signal-to-noise ratio, each time-lapse image was averaged every 30 frames after motion correction using the NoRMCorre software[41]. The calcium signal was extracted using the CaImAn software[42] based on a constrained non-negative matrix factorization algorithm. When several calcium footprints were detected for one cell, we manually combined them. The $\Delta F/F$ value was defined as $(F\text{-}F_0)/F_0$, where F denotes the temporal component values from CaImAn, and $F_0$ denotes the median value of the temporal component. We analyzed signals when mice went to the center (with no walls or black walls) from the closed corridor (with two side black walls), the closed corridor from the center, and the corner (with two side black walls and one side gray wall) from the closed corridor. $Ca^{2+}$ events were also analyzed when mice were exposed to black screens and white screens each for 15 s.

For the $Ca^{2+}$ signal analysis, we selected astrocytic GCaMP6s-positive somas and calculated $\triangle F/F$. To obtain significant $Ca^{2+}$ peaks, we excluded non-significant $Ca^{2+}$ activities within the total signal trace showing less than the mean+SD values of the $\triangle F/F$ activity. Next, we matched the $Ca^{2+}$ activities with VR position and analyzed position-specific (center, corner, and closed) $Ca^{2+}$ activation (number of peaks/ sec). In addition, we defined a behavioral epoch (−5 to +15 s of epoch in which 0 indicates when the mice were entering the center zone in the VR) to precisely analyze two types of position-specific $Ca^{2+}$ responses; i) pre-center-responsive and ii) post-center-responsive cells.

Heat map visualization of two-photon $Ca^{2+}$ results was performed by normalizing the $\triangle F/F$ as follows:

$$a(t)_{norm} = \frac{\Delta F/F(t) - \Delta F/F_{min}}{\Delta F/F_{max} - \Delta F/F_{min}} \qquad (1)$$

in which $a(t)_{norm}$ is within the range of 0–1. To precisely visualize the position-specific $Ca^{2+}$ response, we defined epoch with a range between −5 and 15 s window, indicating that 0 is the time when the mice enter the center position in the VR. Subsequently, we calculated the percentages of position-specific reactive cells, i) from closed to center and ii) from closed to corner or center to closed. Heat map visualization was conducted using MATLAB (2020b, MathWorks, Portola Valley, CA, USA).

**Stereotaxic surgery for the implantation of mono-optic fiber and guide cannula**
To optically manipulate astrocytic activity, two mono-optic fibers were bilaterally implanted in the hippocampus (dorsal, angle: 10°, AP: −1.8, ML: ±2.3, DV: −1.5; ventral, AP: −3.16, ML: 3.25, DV: −3.0 from the bregma) of hGFAP-CreERT2 or wild-type mice. Anchor screws were placed in the skull and fixed with zinc polycarboxylate dental cement. For the administration of pyridoxalphosphate-6-azophenyl-2′,4′-disulfonic acid (PPADS), both a guide cannula (Plastic1, Roanoke, VA, USA) and a mono-optic fiber were implanted in the same region. Following the aforementioned stereotaxic surgery procedure, a guide cannula (C315GMN) covered by its dummy (C315CMN; angle: −20°, AP:

−1.8, ML: 1.25, DV: −1.3 from the bregma) and a mono-optic fiber (angle: −10°, AP: −1.8, ML: 2.3, DV: −1.5 from the bregma) were implanted to target the right hippocampus.

For stereotaxic virus injection, C57BL/6 mice received unilateral injection of 3 μl AVVs-GFAP-ChR2-EYFP (-1 × 10$^{13}$ gene copies (GC)/ml, Korea Institute of Science and Technology, Seoul, Korea) in hippocampus using coordinates AP: −1.8, ML:1.7, DV: −1.6 from the bregma. For optogenetic experiments, optic fibers were unilaterally implanted above the hippocampus 2 weeks after virus injection.

**Immunohistochemistry**
Mice were transcardially perfused with ice-cold 0.1 M phosphate-buffered saline (PBS; pH 7.4) until all blood was removed, followed by perfusion with ice-cold 4% paraformaldehyde in 0.1 M PBS. Whole brains were post-fixed in 4% paraformaldehyde in 0.1 M PBS overnight at 4 °C and cryoprotected with 30% sucrose for 3 days. Coronal 40 μm-thick sections were incubated in cryoprotectant at −20 °C until immunohistochemical staining was performed. The sections were incubated in a blocking solution containing 5% normal donkey serum (Jackson ImmunoResearch, Bar Harbor, ME, USA), 2% BSA (Sigma-Aldrich, St. Louis, MO, USA), and 0.1% Triton X-100 (Sigma-Aldrich) for 1 h at room temperature. Subsequently, the sections were incubated with mouse anti-NeuN (MAB377B, 1:1000; Millipore, Billerica, MA, USA), mouse anti-GFAP (MAB360, 1:1000; Millipore), rabbit anti-S100ß (ab52642, 1:500; Abcam, Cambridge, MA, USA), rabbit anti-Iba1 (019-19,741, 1:1000; Wako, Richmond, VA, USA), and rabbit anti-c-Fos (PC05-100UG, 1:1000; Millipore) antibodies overnight at 4 °C in the blocking solution. After washing with 0.1 M PBS containing 0.1% Triton X-100, the sections were incubated for 1 h with FITC-, Cy3-, or Cy5-conjugated secondary antibodies (1:200, Jackson ImmunoResearch) in the blocking solution at room temperature, washed three times, and then mounted on gelatin-coated glass slides using Vectashield (Vector Laboratories, Inc., Burlingame, CA, USA). Fluorescent images of the mounted sections were obtained using a confocal microscope (LSM800; Carl Zeiss, Jena, Germany).

**Image acquisition and quantification**
Image acquisition was performed using Carl Zeiss confocal laser scanning microscope. Image quantification was performed on images collected using the 20x objective. Laser intensity and gain were kept constant between experiments. The number of GFAP$^+$/S100b$^+$ cells colocalized with EYFP was manually quantified using the Event function of the ZEN 2.0 software within a defined region of interest (ROI) in each tissue section. In addition, quantification of the mean EYFP fluorescence intensity was automatically identified using the ZEN 2.0 software after selection of the ROI. Quantification data were obtained using at least three mice, five tissue sections per mouse, for each experimental group.

**Imaging intracellular astrocyte $Ca^{2+}$ activity in acute hippocampal slices**
Tamoxifen-injected hGFAP-ChR2 mice received a bilateral injection of 2 μl of AVVDJ-GFAP-jRGECO1a (3.97 × 10$^{12}$ GC/ml, IBS Virus Faculty, Institute for Basic Science, Daejeon, Korea) in the hippocampus using the following coordinates from the bregma: AP: −1.8, ML: ± 1.7, DV: −1.6. After 3 weeks, the brains were rapidly removed and placed in an ice-cold, oxygenated (95% O$_2$ and 5% CO$_2$), low-Ca$^{2+}$/high-Mg$^{2+}$ dissection buffer containing 5 mM KCl, 1.23 mM NaH$_2$PO$_4$, 26 mM NaHCO$_3$, 10 mM dextrose, 0.5 mM CaCl$_2$, 10 mM MgCl$_2$, and 212.7 mM sucrose. Transverse hippocampal slices (300 μm) were transferred to a holding chamber in an incubator containing oxygenated (95% O$_2$ and 5% CO$_2$) artificial cerebrospinal fluid (ACSF) composed of 124 mM NaCl, 5 mM KCl, 1.23 mM NaH$_2$PO$_4$, 26 mM NaHCO$_3$, 10 mM dextrose, 2.5 mM CaCl$_2$, and 1.5 mM MgCl$_2$ at 28 °C–30 °C for at least 1 h before imaging. After confirming both ChR2 (EYFP) and jRGECO1a (mCherry)

expressions, jRGECO1a-positive fluorescence activities were imaged using a confocal microscope (LSM800). After imaging basal activities (3 min), a continuous light (473 nm) was delivered for 5 min. Then, the light was turned off and the fluorescent activities were imaged for 2 min. After imaging the fluorescent activities, jRGECO1a-positive cells were selected (ROIs selection) and analyzed using an *EZcalcium*, an open-source toolbox for the analysis of calcium imaging data[43]. Data processing, including motion correction and ROI detection/refinement, was performed according to the developer's instruction. All background signals were corrected, and the obtained data were normalized as Z-score.

## Primary astrocyte culture and in vitro $Ca^{2+}$ assay

Primary astrocytes were isolated from the hippocampus and cortex of postnatal day 1 hGFAP-ChR2 mice and prepared according to an established protocol with minor modifications[44]. Briefly, after the removal of the meninges from the cerebral hemisphere, the tissue was dissociated into a single-cell suspension via gentle repetitive pipetting and then filtered through a 70 μm filter. The cells were cultured in DMEM supplemented with 10 mM HEPES, 10% FBS, 2 mM L-glutamine, and 1x antibiotic/antimycotic in 75 $cm^2$ flasks at 37 °C in a 5% $CO_2$ incubator. The medium was changed every 5 days. After 2 weeks, the flask was shaken at 250 rpm for 2 h at 37 °C, treated with 100 mM L-leucine methyl ester for 60 min to remove microglial cells, and harvested by trypsinization (0.25% trypsin, 0.02% EDTA). After seeding, the cells were incubated with 4-OH-tamoxifen (1 mM, Sigma-Aldrich) for 2 days with a new medium. Calcium responses in the ChR2-expressing astrocytes were measured using single-cell calcium imaging with Rhod-2AM (Invitrogen, Carlsbad, CA, USA). Primary cells were plated on PDL-coated cover glasses and incubated overnight. The cells were incubated for 50 min at 37 °C with 2 μM Rhod-2AM in HBSS containing 25 mM HEPES (pH 7.5) and washed with HBSS−HEPES twice before assays. A baseline reading was performed for 100 s before blue light stimulation. The intracellular calcium level was measured via digital video microfluorometry using an intensified charge-coupled device camera (Cascade; Roper Scientific, Trenton, NJ, USA) connected to a microscope and analyzed using the MetaFluor software (Universal Imaging Corp., Downingtown, PA, USA). Increase in fluorescence intensity over baseline was calculated for each trace and reported as ΔF/F0.

## Optogenetic astrocyte stimulation

For optogenetic manipulation of hippocampal astrocytes, a mono-optic fiber was implanted in the hippocampus of hGFAP-ChR2 transgenic mice. For bilateral stimulation, 200 μm optic fibers (FP200URT; Thorlabs Inc., Newton, NJ, USA) with a 1.25 mm ceramic ferrule (CFLC230-10; Thorlabs Inc.) were used. The mice were stimulated with 473 nm blue light (1–3 mW) delivered for 5 min continuously (Shanghai Laser & Optic Century, Shanghai, China) during in vivo or ex vivo experiments. For calcium imaging experiments, the ChR2-expressing astrocyte received light stimulation for 20 s.

## Electrophysiology

Transverse hippocampal slices (300 μm) were prepared. Briefly, the mice were anesthetized with isoflurane and decapitated. The brains were rapidly removed and placed in ice-cold, oxygenated (95% $O_2$ and 5% $CO_2$), low-$Ca^{2+}$/high-$Mg^{2+}$ dissection buffer containing 5 mM KCl, 1.23 mM $NaH_2PO_4$, 26 mM $NaHCO_3$, 10 mM dextrose, 0.5 mM $CaCl_2$, 10 mM $MgCl_2$, and 212.7 mM sucrose. Slices were transferred to a holding chamber in an incubator containing oxygenated (95% $O_2$ and 5% $CO_2$) ACSF composed of 124 mM NaCl, 5 mM KCl, 1.23 mM $NaH_2PO_4$, 26 mM $NaHCO_3$, 10 mM dextrose, 2.5 mM $CaCl_2$, and 1.5 mM $MgCl_2$ at 28 °C–30 °C for at least 1 h before recording. After recovery, the slices were transferred to a recording chamber and perfused continuously with ACSF perfused with 95% $O_2$ and 5% $CO_2$ at a flow rate

of 2 ml/min. The slices were equilibrated for 5 min prior to the recordings, and all experiments were conducted at 28 °C–30 °C. Recordings were obtained using a MultiClamp 700B amplifier (Molecular Devices, Sunnyvale, CA, USA) under visual control with differential interference contrast illumination on an upright microscope (BX51WI; Olympus). Patch pipettes (5–7 MΩ) were filled with 135 mM K-gluconate, 8 mM NaCl, 10 mM HEPES, 2 mM ATP-Na, and 0.2 mM GTP-Na to record sEPSCs in the voltage-clamp mode and excitability in the current-clamp mode (pH 7.4 and 280–290 mOsm). Only cells with access resistance <20 MΩ and input resistance >100 MΩ were evaluated.

The sEPSC experiment was conducted within a 10 min scheme (3 min without light for basal, 5 min with light stimulation, and 2 min without light) for each cell. Basal sEPSC was recorded for the first 3 min without light stimulation (0–3 min). Then, 473 nm light (continuous) was delivered to the hippocampal slice through an optic fiber in a patch pipette for 5 min (3–8 min), and the light was turned off for the last 2 min of recording (8–10 min). The extracellular recording solution consisted of ACSF supplemented with picrotoxin (100 μM). An identical experimental procedure (light-off for 3 min, light-on for 5 min, and light-off for 2 min) was used for the sEPSC recording with optogenetic light stimulation in the presence of PPADS (Tocris Bioscience, Bristol, UK). The extracellular recording solution consisted of ACSF supplemented with picrotoxin (100 μM) and PPADS (100 μM).

The sEPSC experiment with two times optogenetic light stimulation was conducted within a 20 min scheme. Basal sEPSC was recorded for the first 3 min without light stimulation (0–3 min). Then, 473 nm light (continuous) was delivered to the hippocampal slice through an optic fiber in a patch pipette for 5 min (3–8 min), and the light was turned off for the last 2 min of recording (8–10 min). For the second light stimulation, 473 nm light (continuous) was delivered to the hippocampal slice through an optic fiber in a patch pipette for 5 min (10–15 min), and the light was turned off for the last 2 min of recording (15–18 min). The extracellular recording solution consisted of ACSF supplemented with picrotoxin (100 μM).

For the sEPSC recording while changing extracellular $K^+$ concentration, ACSF solutions with different KCl concentrations were prepared (5 mM, 7 mM, or 9 mM KCl). Basal sEPSC was recorded for 10 min with 5 mM KCl ACSF buffer, and then the buffer was continuously changed (flow rate of 2 ml/min) into ACSF with 7 mM or 9 mM KCl during the recording. After the 10 min recording in 7 mM or 9 mM KCl-containing ACSF, the buffer was changed back to 5 mM KCl-containing ACSF, and sEPSC was recorded for 10 min. The entire recording time was 30 min for each cell. The extracellular recording solution consisted of ACSF supplemented with picrotoxin (100 μM).

For the sEPSC recording while changing extracellular ATP concentration, ACSF solutions with 100 μM of ATP were prepared. Basal sEPSC was recorded for 10 min without ATP in ACSF buffer, and then the buffer was continuously changed (flow rate of 2 ml/min) into ACSF with 100 μM of ATP. After the 10 min recording in ATP-containing ACSF, the buffer was changed back to normal ACSF, and sEPSC was recorded for 10 min. The entire recording time was 30 min for each cell. The extracellular recording solution consisted of ACSF supplemented with picrotoxin (100 μM).

We recorded neuronal excitability with current clamp mode in response to injection of a current ramp from −50 to 150 pA with 5 s duration. All electrophysiology data were acquired and analyzed using pClamp 10.5 (Molecular Devices). Signals were filtered at 2 kHz and digitized at 10 kHz using Digidata 1550 A (Axon Instruments, Union City, CA, USA).

## Behavioral studies

Each mouse was handled for 5 min every day for 5 days before conducting behavioral assessments. On all behavioral testing days, the animals were moved to the test room and left for acclimatization for at

least 30 min. The behavior tests comprised the elevated plus maze (EPM), open field test (OFT), T-maze, and real-time place preference (RTPP) and were monitored using a computerized tracking system (SMART 3.0, Panlab Harvard Apparatus, Holliston, MA, USA). All the behavioral testing and analysis were conducted blind to the experimental conditions.

## EPM

The behavioral apparatus consisted of two open arms and two closed arms (width 5 cm × length 30 cm) elevated 50 cm above the floor and dimly illuminated (300 lux). The mice were placed individually in the center of the maze facing an open arm and allowed to freely explore for 5 min. hGFAP-ChR2 mice were exposed to the EPM apparatus for 15 min, with 5 min light-off and 5 min light-on epochs, followed by a second 5 min light-off epoch. The time spent or distance traveled in each arm was analyzed using a video tracking system. The maze was cleaned with 70% ethanol after each test to prevent olfactory influence from the previously tested mouse.

## OFT

The OFT apparatus consisted of a brightly illuminated 40 × 40 cm square arena surrounded by a 40 cm-high wall. The mice were individually placed in the center of the arena, and their locomotion activity was monitored using an automatic system for 5 min (C57BL/6), 15 min, or 1 h (hGFAP-ChR2). In the hGFAP-ChR2 mice, light stimulation of 473 nm (continuation or 20 Hz) was turned on 5 min after starting the session and turned off after 5 min. The distance traveled and time spent in each area were analyzed using an automated video tracking system. The activity chamber was cleaned with 70% ethanol after each use to eliminate any olfactory cues from the previously tested mouse.

## T-maze

The T-shaped elevated maze contained one start arm (30 × 10 cm) and two goal arms (30 × 10 cm, new and old arms, with the arms counterbalanced) with black stripes or gray circles. During the training session, mice were individually placed in the start arm facing the wall, with one of the goal arms (new arm) blocked, and they were allowed to freely explore the maze for 20 min. The hGFAP-ChR2 mice received 5 min of light stimulation at the beginning of the training session. The retention test was conducted 5 min after training. Before the test session, the block was removed, and the mice were placed at the end of the start arm facing the wall and allowed to freely explore for 5 min. The percentage of time spent in the new arm and the total exploration time were measured using an automated video tracking system. New arm preference was calculated by dividing the time spent in the new arm by the time spent in both goal arms (new and old).

## RTPP

RTPP was performed in a rectangular apparatus consisting of two side chambers (25 × 25 × 30 cm each) with black stripes and gray circles on the walls, connected by a center corridor (10 × 10 cm). The RTPP test consisted of three phases of behavioral testing over 3 days. At the beginning of the experiment, the mice were placed in the center chamber and allowed to freely explore the entire apparatus. Entry to the stimulated chamber triggered light stimulation. On day 1, individual mice were placed in the center chamber and allowed to freely explore the entire apparatus for 15 min (habituation). On day 2, one of the side chambers was exposed to light stimulation, and the other chamber was not. Each time the mice crossed the light-stimulated chamber, light was activated until the mouse crossed back to the center corridor, for 20 min maximum. On day 3, the mice were again allowed to freely explore the entire apparatus for 15 min. The preference of chamber was calculated as percentage of the time spent in the stimulation chamber over the total time spent in both chambers.

All behaviors were recorded using an automated video tracking system.

## EPM and OFT with PPADS treatment

The mice were stereotactically implanted with an optic fiber and a guide cannula into the hippocampal region as described above. Three days after the surgery, EPM or OFT was performed 30 min after drug delivery via the guide cannula using 1 µl of PPADS (100 µM) at a rate of 0.1 µl/min. Each behavior was tested at least 48 h after the previous test to minimize potential stress effects.

## Gliotransmitter measurement

**Extracellular ATP measurement.** The ATP level was determined using the ATP Bioluminescence Assay Kit (Sigma-Aldrich) based on the manufacturer instruction[45,46]. FPL 67156 (ARL 67156, Sigma-Aldrich), an ectonucleotidase inhibitor 6-N,N-diethyl-β-γ-dibromomethylene-d-denosine-5-triphosphate, was added to the ACSF or to the conditioned medium throughout the experiment to minimize ATP hydrolysis. For the ATP release measurements, the hippocampal acute slices or primary cultured astrocytes were incubated in oxygenated ACSF or conditioned medium for 10 min with light stimulation. The ACSF or medium was collected and transferred to 96-well plates. Luciferase activity was measured using a luminometer (SPARK 10 M, Tecan, Grödig, Austria) according to the manufacturer's protocols and normalized to the total protein level of each sample.

**Glutamate and D-serine assay.** For glutamate and D-serine assays, acute hippocampus slices were collected from hGFAP-ChR2 transgenic mice and light stimulated for 10 min. All slices were incubated in oxygenated ACSF with light stimulation, and the ACSF was collected. All assays were conducted using a 96-well microplate immediately after sample collection. Glutamate was measured using a colorimetric assay kit (#K629-100; BioVision, Milpitas, CA, USA) and D-serine using a fluorometric assay kit (#K743-100; BioVision) according to the manufacturer's protocols. The values were normalized to the total protein of each sample.

## Statistics and reproducibility

Statistical significance was determined using one-way analysis of variance (ANOVA) and two-way repeated measure ANOVA followed by *post hoc* Bonferroni and LSD tests for multiple comparisons. For comparisons between two groups, two-tails Mann–Whitney test, independent two sample *t*-test, and paired *t*-test were used. The Kolmogorov–Smirnov test was used to analyze cumulative probability distributions. All data are expressed as mean±standard error of the mean (SEM), and differences were considered statistically significant when the *p*-value was <0.05. Detailed statistics are provided in Supplementary Data.

## Reporting summary

Further information on research design is available in the Nature Research Reporting Summary linked to this article.

## Data availability

All data generated in this study are provided in the Supplementary Data/Source Data file. The raw data images that support the findings of the current study are available from the corresponding author upon reasonable request. Source data are provided with this paper.

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

## Acknowledgements

This research was funded by the Samsung Science & Technology Foundation (SSTF-BA1502-13) and the National Research Foundation (NRF-2016R1A6A3A11931502 and 2021R1A4A1021594).

## Author contributions

W.-H.C. and S.J. L. conceived the project, designed experiments, prepared the manuscript. K.N. performed electrophysiological recordings and analyzed electrophysiological data. W.-H.C., B.H.L. designed, engineered and performed calcium imaging experiments and analyzed data. E.B. performomed histology. W.-H.C., K.N., S.B.J., H.Y. P., S.J. L. wrote and refined the article.

## Competing interests

The authors declare no competing interests.
