## [Peer Review File · Nature Communications]

Hippocampal astrocytes modulate anxiety-like behaviorREVIEWER COMMENTS

Reviewer #1 (Remarks to the Author):

Cho et al. investigated how hippocampal astrocytes contribute to synaptic function that underlies anxiety behavior. The authors used conditional hGFAP-CreERT2 mouse line to target astrocytes in adult mice and found that astrocyte calcium activity appears to synchronize with anxiety states in a virtual reality environment with two-photon in vivo imaging. The authors further expressed channelrhodopsin ChR2 in astrocytes globally with hGFAP-CreERT2 mice and identified synaptic alterations presumably through ATP signaling in the hippocampus.

Increasing evidence has demonstrated the importance of astrocytes in regulating neural circuits and animal behavior. Yet, astrocytic involvement in anxiety behavior is largely unexplored. Thus, the work by Cho et al. is novel from this perspective. However, some of the conclusions drawn from the data are open to interpretation and there are several major issues that the authors need to adequately address. Without further supporting evidence, the current conclusions are unconvincing.

Major points:

1. Inducible hGFAP-CreERT2 mouse line was used to drive the expression of calcium indicator GCaMP6s or channelrhodopsin in astrocytes globally throughout the paper. Although tamoxifen was given in adulthood (two-month old of age), hGFAP promoter is known to target neurons in multiple brain areas including dentate gyrus in the hippocampus. This is mainly because GFAP is highly expressed in neuronal progenitor cells, which are present in the dentate gyrus in adulthood. Therefore, in addition to the expression efficiency shown in Fig 2b and Supplementary Fig 1b as well as the expression intensity shown in Supplementary Fig3, the authors should provide detailed analysis of the expression specificity. Given the reliance upon the use of the hGFAP-CreERT2 mouse line, quantifications of percentages of positive cells expressing GCaMP6s or ChR2 that colocalize with NeuN+ cells from different brain regions as well as various hippocampal regions, i.e. CA1-3 and dentate gyrus, are warranted.
2. In Fig 1, synchronized calcium activities were detected in astrocytes when transitioning from the closed arm to the open center in the VR environment. The authors interpreted the data as “the $[Ca^{2+}]_i$ activity of hippocampal astrocytes discerns or reflects anxiety state”. However, another explanation is that the designed VR does not necessarily create an anxiogenic environment for the mice and therefore the calcium activity of hippocampal astrocytes is purely responsive to changes of the visual stimuli such as images, lighting, shapes etc from the screen.
3. ChR2 was used by the authors to “activate” astrocytes in the hippocampus. However, it is important to note that astrocytes are not electrically excitable cells, i.e. they have few if any voltage-gated ion channels and cannot propagate voltage signals as neurons do. This makes the application of ChR2 to “excite” or “activate” astrocytes controversial. As the authors already pointed out, a previous study by Octeau et al. demonstrated that ChR2 activation in astrocytes triggers a significant

elevation of extracellular K⁺ that is sufficient to change neuronal excitability and cFos expression. This K⁺ model can explain the prolonged effects on behaviors as well as cFos increase reported in Fig 2 and Fig 3 of the current work. The data presented in Supplementary Fig 9 by elevating extracellular K⁺ concentration in brain slices cannot completely rule out this possibility.

4. It's interesting that both dorsal and ventral stimulation of ChR2 triggered neuronal changes in the entire hippocampus as the penetration of blue light is limited in the tissue. The authors should elaborate more on this issue to rule out possibilities, for instance, long-range neuronal projection effects due to ChR2 contamination. Local injection of AAVs to drive ChR2 expression in hippocampal astrocytes could be one way to tease this apart.

Minor points:

1. The cortex above the hippocampus was aspirated to facilitate in vivo imaging. However, astrocytes may become reactive and change calcium signals due to the invasive surgical procedure, this possibility should be tested with astrocyte reactive marker staining or morphological analysis.

2. It's unclear the exact depth or volume where the in vivo calcium imaging was performed at in Fig 1. From Fig 1d, the circled area covers a relatively large region in between CA1 and dentate gyrus, but the representative image in Fig 1e seems to be acquired from one single focal plane. The authors should provide additional details to clarify this point.

3. In Fig 2c, ChR2 activation induced calcium elevation in primary cultured astrocytes collected from P1 mice. This experiment should be repeated in vivo in adult mice at matching age with other behavioral and imaging evaluations.

4. The same data were presented in Fig 2f and 2g, although in slightly different formats. The authors should combine both analyses or move one analysis to the Supplementary information.

Reviewer #2 (Remarks to the Author):

In this paper, Choi et al describe a role for hippocampal astrocytes in anxiety-related behaviors. Using 2P imaging they first describe that hippocampal astrocytes seem to be activated around a VR maze exploration. They then manipulate astrocytes activity using a ChR2 genetic construct and show that light-induced activation of astrocytes induces anxiolytic effects and an increase in exploratory behavior. Finally, they found that astrocytes stimulation leads to ATP release that is necessary to the above-mentioned functions.

The findings are very interesting. Yet, the manuscript suffers some overstatement (for example “astrocytes influence affective state have not been elucidated” is not entirely right, as at least one recent publication describes a role for astrocytes in anxiety (Wahis et al., Nature Neuroscience, 2021). This is related to another easy-to-correct weakness of the manuscript, its apparent lack of literature basement. Indeed, authors do cite only a dozens of articles, allowing some introduction of method statement unjustified – that should be corrected. For example, the protocol used for VR is difficult to understand (what is closed / center / corner? What is the rationale behind restricting the VR movement of animal in 1D with a needle?) and not strengthened by literature. All in all, while authors should dampen their conclusions and cite more scientific article to justify their hypothesis and methods, I believe the overall message of the manuscript is of strong interest for Nature Communication. Also, in its current form the manuscript dos not have any discussion - this needs to be added.

In addition, I have a number of comments that I list above, by order of appearance:

- In most figures the Y axis are often missing, making the figure difficult to read.
- Statistics: Most of the time, t-tests are not appropriate, author must use multiple comparison tests such as ANOVA, after testing the normal distribution of the data. This is particularly true for behavior data.
- Bibliography: authors should add much more references to the manuscript in order to strengthen their hypothesis and methods. This might be true for both methods and introductions. Particularly, ChR2 in astrocytes is rarely used – note that the K⁺ control in Fig S9 is welcome and very important, confirming that such opsin might be a good tool to study astrocytes.
- Fig 1: how many animals and cells recorded? Looking at the figure it seems the correlation between hippocampal astrocytes activity and anxiogenic environment exposure is very weak.
- Fig 1g: quantification is missing for “center” part of the VR lab., and comparisons should be performed using multiple comparisons tests such as ANOVA if a normal distribution of the data is tested.
- Fig S1: how many cells counted? % ok, n animals ok, but n cells?
- Fig S2: bar graph show that the analysis was performed on only ~30 cells? It is very low according to the statement authors made in the text that 2 populations of astrocytes exst, one pre responsive and one post responsive. How authors exclude that those are not spontaneous random events? This is an example of overstatement.
- Fig S3. g and h panels need quantifications. How many EYFP cells were NeuN+? GFAP+? How many GFAP+ were EYFP+? That to assess both efficiency and specificity of the construct. The remarks are also valuable for Fig S1 construct.
- Fig 2. Because authors used full transgenic mice, the placement of the optic fibers to stimulate astrocytes is critical. Can they provide pictures allowing the reader to assess the right localization of such fibers?

- Fig S4. Most of the y axis labels are missing.

- Fig 2d: Accordingly, to the figure and the methods the light stimulation was 5min continuous. This seems to me an enormous stimulation, that should probably be deleterious for the cells, eventually killing some of them. Did control animals also receive 5min blue light? Do the authors verify, at least in ex vivo slices, that such stimulation does not kill the astrocytes or neighboring neurons? Methods say the 5min continuous was also applied ex vivo, however panel c shows a ~20s stimulation. Such deleterious effect might explain the long lasting behavioral effects they see, and the absence of "washout" effect (i.e. "after" BL stimulation) (while the OFT data seem to be a control in themselves).

- Fig 3. Could authors provide a quantification of the c-fos increase?

- Fig 3: What is the PPADS concentration the authors used? How do authors explain the apparent discrepancy between the organotypic slice experiments, showing that ATP is the only gliotransmitter to display an increase in concentration, and the persistence of behavioral effects after PPADS use? One lead might be that PPADS do not block all purinergic receptors. Interesting hypothesis might be the involvement of other receptors, such as A1 or A2A which can be blocked with CPT or SCH 58261.

- Line 96 the appeal to fig S3h seems misplaced

Reviewer #3 (Remarks to the Author):

The manuscript entitled "Hippocampal astrocytes modulate anxiety" by Woo-Hyaun Cho et al. provides evidence for the involvement of astrocytes in regulating the animal response to the anxiogenic environment. It is shown that upon an increase in intracellular calcium concentration, astrocytes release ATP which in turn increases the sEPSC frequency in neighboring neurons, astrocytes in both the ventral and dorsal hippocampus participate in the modulation of anxiety behaviors. Using a combination of in vivo and in vitro approaches the authors present a cellular mechanism by which astrocyte-neuron communication affects brain activity and behavior. Overall this study is interesting and timely, however, there are several major and minor concerns that authors should address before being considered for publication.

1) The manuscript can be overall improved by better describing and discussing the reported results. Many figure panels are not even cited in the text whereas many others are described all together with one sentence making it hard for the reader to follow the story and to interpret the reported data.

2) Figure 1 and Supp.:

- A GRIN lens implant causes a big amount of brain damage. While I understand the need for this approach, it would be good to control for reactive astrocytes and consider this technical limitation

when describing the results. Moreover, in the method section, there is no mention of the GRIN lens, but rather of a stainless-steel cannula attached to a coverslip. Please comment and adjust.

- What is the meaning of measuring the amplitude of calcium peak? This is not a ratiometric calcium indicator and the amplitude can be different for many different reasons that can be independent of the calcium concentration.
- It is described that there are no cells active in the closed arms, but from the heat map this does not seem to be the case. From cell-40 to the end there are cells active since the beginning of the peri-event histogram (-3s). Also, the cells grouped as “pre-responsive” to entering the center are active in the middle of the closed arm and represent 50% of the population reported in both conditions (panel I). Please explain why these are not considered as active in the closed arm instead.
- The tracking ball data (i.e., distance traveled, speed) and the reward release time should be reported to make sure the astrocytes are not active in correlation to other variables.
- The analysis of the calcium activity has been limited to the cell soma, what is the degree of calcium activity in the main branches and microdomains? Is there an increase in relation to the mouse position in the VR environment, distance traveled, speed or reward?
- At line 74 Supp figure 1 is not described.
- The legend of Fig Supp figure 1 seems to be wrong for panel a. The Chr2 expressing astrocytes are described, but they are GCaMP6s expressing astrocytes.
- The whole analysis in Supp figure 2 is not clear and not described in the main text. From the legend, it seems like different time windows were used to define cells compared to the one reported in Fig1. Please explain.

3) Figure2 and Supp:

- What is the effect of Chr2 stimulation on the calcium activity of astrocytes in vivo? Does the optogenetic-induced calcium activity last longer than the behaviorally-elicited activity?
- Although reported as an important effect throughout the paper, the long-lasting behavioral change in the second 5min after the optogenetic stimulation is not further discussed. What is the possible reason?
- Since panels 2h-I describe an increased mean speed upon optogenetic stimulation of astrocytes, I even strongly recommend analyzing the mean speed in the VR experiments and see if there is a correlation between calcium activity and traveling speed.
- Do the neurons in both the dorsal and ventral parts of the hippocampus respond in the same way to the astrocytes manipulation?
- How do you explain the results in Supp figure 5? What is the percentage of time spent in the center versus the sides of the open field arena?
- It is not clear to me how the elevated calcium activity reported in figure 1 in response to an anxiogenic environment correlates to the anxiolytic effect of elicited calcium activity reported in figure 2. Please clarify.
- From the example reported in panel j, it does not seem like the hGFAP-ChR2 mice are spending more time in the center compared to controls.

- In Supp figure 6 please compare the %of time spent in the light chamber also in the session before and after the light stimulation. The n should also increase as in Supp figure 6d, it seems that 3/5 mice are behaving in opposite ways (CTRL spending more time in the stim side and hGFAP-ChR2 spending less time in the stim chamber) and the remaining 2 mice do the opposite. Moreover, no statistical tests are reported in the legend.

4) Figure 3 and Supp:

- How does the time scale of the sEPSCs increased frequency fit the behavioral effects lasting for an extra 5min after the astrocyte stimulation?
- The experiments reported in Supp fig.9 are based on a study in which is described the increase in extracellular K⁺ level upon astrocytes optogenetic excitation in the striatum. However, the maximum reported concentration of K⁺ (7.5 mM) is based on 1000 light pulses of 25ms, which is 25s light stimulation, without considering an inter-pulse-interval. In the present manuscript, the stimulation time is 5 min with continuous light. To compare the data reported here with the previous study there should be first an understanding of what is the maximum concentration of extracellular K⁺ after 5 min stimulation and then study the effect of that concentration on frequency and amplitude of sEPSC.
- Does the neuronal firing rate increase with astrocyte optogenetic stimulation alone?
- The light stimulation for ATP, Glutamate, and D/DL-serine measurement is twice the one used in vivo. Please explain why.
- In panel f the cumulative probability looks significantly different even though the mean is not. Please test and report.
- Also in this case, in Supp figure 10 there is a major point about the increased speed and total travel distance which is not blocked by PPADS injection.

5) All cumulative probability distribution should be analyzed using the Kolmogorov-Smirnov test. The mean frequency analysis is not sufficient as the distributions in many cases look different even though the mean is not.

6) It is not always clear which statistical test has been performed on the data. For some analysis (i.e., Figure 2 and in Figure 3) it is necessary to know what test has been used to define statistical significance. Due to the presence of many graphs, it becomes difficult to know for sure by reading the legend and the statistical methods.

7) The t-value for all the t-test and the F value for all the ANOVA need to be reported.

Reviewer #1 (Remarks to the Author):

Cho et al. investigated how hippocampal astrocytes contribute to synaptic function that underlies anxiety behavior. The authors used conditional hGFAP-CreERT2 mouse line to target astrocytes in adult mice and found that astrocyte calcium activity appears to synchronize with anxiety states in a virtual reality environment with two-photon in vivo imaging. The authors further expressed channelrhodopsin ChR2 in astrocytes globally with hGFAP-CreERT2 mice and identified synaptic alterations presumably through ATP signaling in the hippocampus.

Increasing evidence has demonstrated the importance of astrocytes in regulating neural circuits and animal behavior. Yet, astrocytic involvement in anxiety behavior is largely unexplored. Thus, the work by Cho et al. is novel from this perspective. However, some of the conclusions drawn from the data are open to interpretation and there are several major issues that the authors need to adequately address. Without further supporting evidence, the current conclusions are unconvincing.

Major points:

1. Inducible hGFAP-CreERT2 mouse line was used to drive the expression of calcium indicator GCaMP6s or channelrhodopsin in astrocytes globally throughout the paper. Although tamoxifen was given in adulthood (two-month old of age), hGFAP promoter is known to target neurons in multiple brain areas including dentate gyrus in the hippocampus. This is mainly because GFAP is highly expressed in neuronal progenitor cells, which are present in the dentate gyrus in adulthood. Therefore, in addition to the expression efficiency shown in Fig 2b and Supplementary Fig 1b as well as the expression intensity shown in Supplementary Fig3, the authors should provide detailed analysis of the expression specificity. Given the reliance upon

the use of the hGFAP-CreERT2 mouse line, quantifications of percentages of positive cells expressing GCaMP6s or ChR2 that colocalize with NeuN+ cells from different brain regions as well as various hippocampal regions, i.e. CA1-3 and dentate gyrus, are warranted.

Response :

We appreciated the reviewer for their helpful comment. As suggested, we have analyzed more thoroughly GCaMP6s- and ChR2-expressing cell types upon tamoxifen injection. However, we could not detect any colocalization of GCaMP6 or ChR2 signals in NeuN-positive neurons in any brain sub-area including dentate gyrus, and CA1 and CA3 of the hippocampus. We updated these data and included them in Supplementary figure 1a, c and Supplementary figure 5g, h of the revised manuscript.

2. In Fig 1, synchronized calcium activities were detected in astrocytes when transitioning from the closed arm to the open center in the VR environment. The authors interpreted the data as “the $[Ca^{2+}]_i$ activity of hippocampal astrocytes discerns or reflects anxiety state”. However, another explanation is that the designed VR does not necessarily create an anxiogenic environment for the mice and therefore the calcium activity of hippocampal astrocytes is purely responsive to changes of the visual stimuli such as images, lighting, shapes etc from the screen.

Response :

Elevated plus maze (EPM) is a standard behavioral test to measure mouse anxiety¹, which we recapitulated in virtual reality (VR). We designed a VR screen as if the mouse were to enter a bright and open center through a dark walled corridor. When we measured the time spent in the center vs. corner or closed zones in VR, it was much shorter in the center zone compared to corner or closed zones (Supplementary figure 1e). This is reminiscent to the mouse behavior in real EPM test and suggest that mice feel anxious in open zone even in virtual reality.

To exclude the possibility that hippocampal astrocytes respond to different visual images, we changed the image of the bright white open center for an image of a black-walled corridor without star that was not supposed to trigger anxiety. In this VR setting, the number of Ca²⁺ peaks was not different between the center and the closed zone even though mice were exposed to different images (Supplementary fig 4b,c). These results support that hippocampal astrocytes respond not merely to the different images, but most likely to the anxiogenic environment. We included these new data in Supplementary figure 4 of the revised manuscript.

To further confirm that hippocampal astrocytes are responding to anxiogenic stimuli, we subjected mice to a fear conditioning experiments, another type of anxiogenic environment. First, mice were presented with a tone for 20 seconds and then given a foot shock for the last 2 seconds which was repeated twice with an interval of 2 minutes. After conditioning, hippocampal astrocyte Ca²⁺ events were monitored by two-photon imaging with the conditioned tone. During acoustic tone, astrocyte Ca²⁺ events increased and lasted even after turning off the acoustic tone. These results suggest that hippocampal astrocyte calcium activities are involved in other types of anxiogenic stimuli such as fear-conditioned acoustic tone. These data support that hippocampal astrocytes respond not merely to the bright and open environment but indeed to anxiogenic stimuli. The data of fear-conditioning experiment were attached to this response letter to the reviewers' comments as Reference Figure 1 below.

Reference Figure 1. **a**, Fear conditioning. **b**, Heat map trace of normalization of GCaMP6s signals. **c**, Number of Ca^{2+} peak (left) and peak duration (right). Left, One-way ANOVA, $F(2,81) = 9.076$, $***p = 0.001$, *post hoc* LSD, pre vs during, $p = 0.081$, during vs post, $*p = 0.015$, pre vs post, $***p = 0.000$. Right, One-way ANOVA, $F(2,81) = 29.398$, $***p = 0.001$, *post hoc* LSD, pre vs during, $p = 0.066$, during vs post, $***p = 0.001$, pre vs post, $***p = 0.001$.

3. ChR2 was used by the authors to “activate” astrocytes in the hippocampus. However, it is important to note that astrocytes are not electrically excitable cells, i.e. they have few if any voltage-gated ion channels and cannot propagate voltage signals as neurons do. This makes the application of ChR2 to “excite” or “activate” astrocytes controversial. As the authors already pointed out, a previous study by Oceau et al. demonstrated that ChR2 activation in astrocytes triggers a significant elevation of extracellular K^+ that is sufficient to change neuronal excitability and cFos expression. This K^+ model can explain the prolonged effects on behaviors as well as cFos increase reported in Fig 2 and Fig 3 of the current work. The data presented in Supplementary Fig 9 by elevating extracellular K^+ concentration in brain slices cannot completely rule out this possibility.

Response :

We appreciate the reviewer's comment. When we mentioned “astrocyte activation” in this study, we did not mean “electric excitation”. Rather, we meant intracellular astrocyte calcium activation and subsequent gliotransmitter release. ChR2 expression followed by light stimulation was utilized to induce astrocyte calcium signaling. Astrocyte calcium signaling induction by optogenetic ChR2 stimulation was reported by other studies^{2, 3}, and also confirmed in our study (Fig. 2c). In addition, many studies previously utilized this method to induce astrocyte calcium signaling and thereby study the role of astrocyte activation²⁻⁴.

To address reviewer's concern, we have performed additional electrophysiological experiments to assess extracellular K^+ effect on DG granule cells. We measured sEPSC of DG granule cells while increasing K^+ concentration in ACSF buffer to either 7 or 9 mM. In these experiments, both sEPSC amplitude and frequency were increased in 9 mM K^+ concentration (Supplementary fig. 11 k and l), but not in 7 mM (Supplementary fig. 11 g and h). Such electrophysiological changes by extracellular K^+ increase does not exactly match with the

electrophysiological impact of optogenetic astrocyte stimulation, by which only sEPSC frequency was increased but not amplitude (Figure 3e-g). We included these new data in Supplementary fig. 11 of the revised manuscript.

Moreover, the electrophysiological changes as well as the anxiety-related behavioral changes were rescued by PPADS, a purinergic receptor antagonist. Therefore, although we cannot completely exclude the impact of extracellular K^+ increase by optogenetic astrocytes stimulation, we would like to argue that anxiety-like behaviors induced by optogenetic hippocampal astrocytes stimulation are mediated by astrocyte-derived gliotransmitter, ATP.

4. It's interesting that both dorsal and ventral stimulation of ChR2 triggered neuronal changes in the entire hippocampus as the penetration of blue light is limited in the tissue. The authors should elaborate more on this issue to rule out possibilities, for instance, long-range neuronal projection effects due to ChR2 contamination. Local injection of AAVs to drive ChR2 expression in hippocampal astrocytes could be one way to tease this apart.

Response :

We have performed the additional experiment the reviewer suggested. We unilaterally injected AAVs-GFAP-ChR2-EYFP virus to the dorsal hippocampus and then stimulated with blue light. Similar to the data using transgenic mice, c-Fos was robustly expressed in DG, CA1 and CA3 of dorsal and ventral hippocampus. However, there was no c-Fos induction in the contralateral hippocampus. These data argue against the possibility of long-range neuronal projection effects via ChR2 contamination. We included these new data as Supplementary fig. 9c in the revised manuscript.

Minor points:

1. The cortex above the hippocampus was aspirated to facilitate in vivo imaging. However, astrocytes may become reactive and change calcium signals due to the invasive surgical procedure, this possibility should be tested with astrocyte reactive marker staining or morphological analysis.

Response :

We measured Ca^{2+} activity of hippocampal astrocytes that locate between CA1 stratum pyramidale and molecular layer of the dentate gyrus. Upon microsurgery to make cranial window, astrogliosis or reactive astrocytes were detected only in stratum oriens or pyramidale layer, the uppermost layer of the hippocampus. However, in stratum radiatum and stratum lacunosum layers, where astrocyte Ca^{2+} events were measured, there was no significant morphological activation of astrocytes. We included these additional data as Supplementary figure 1f in the revised manuscript.

2. It's unclear the exact depth or volume where the in vivo calcium imaging was performed at in Fig 1. From Fig 1d, the circled area covers a relatively large region in between CA1 and dentate gyrus, but the representative image in Fig 1e seems to be acquired from one single focal plane. The authors should provide additional details to clarify this point.

Response :

As mentioned above, we measured Ca^{2+} activity of hippocampal astrocytes located between CA1 stratum pyramidale and molecular layer of the dentate gyrus. To be specific, we targeted cells at a depth of 50~200 μm based on the CA1 pyramidal layer. This is specified in the Supplementary materials and method of the revised manuscript.

3. In Fig 2c, ChR2 activation induced calcium elevation in primary cultured astrocytes

collected from P1 mice. This experiment should be repeated in vivo in adult mice at matching age with other behavioral and imaging evaluations.

Response :

Astrocyte calcium activation upon ChR2 stimulation in adult mouse hippocampus has been already reported⁵. In behaving mouse in vivo, the excitation wavelength we used overlapped with calcium signal in GCaM6s transgenic mice, thus it was not technically feasible to detect astrocyte calcium signal while optogenetically activating astrocytes, which we hope the reviewer understand.

4. The same data were presented in Fig 2f and 2g, although in slightly different formats. The authors should combine both analyses or move one analysis to the Supplementary information.

Response :

Figure 2g and other graphs with the same issue were moved to Supplementary information as suggested.

Reviewer #2 (Remarks to the Author):

In this paper, Choi et al describe a role for hippocampal astrocytes in anxiety-related behaviors. Using 2P imaging they first describe that hippocampal astrocytes seem to be activated around a VR maze exploration. They then manipulate astrocytes activity using a ChR2 genetic construct and show that light-induced activation of astrocytes induces anxiolytic effects and an increase in exploratory behavior. Finally, they found that astrocytes stimulation leads to ATP release that is necessary to the above-mentioned functions.

The findings are very interesting. Yet, the manuscript suffers some overstatement (for example “astrocytes influence affective state have not been elucidated” is not entirely right, as at least one recent publication describes a role for astrocytes in anxiety (Wahis et al., Nature Neuroscience, 2021). This is related to another easy-to-correct weakness of the manuscript, its apparent lack of literature basement. Indeed, authors do cite only a dozens of articles, allowing some introduction of method statement unjustified – that should be corrected. For example, the protocol used for VR is difficult to understand (what is closed / center / corner? What is the rationale behind restricting the VR movement of animal in 1D with a needle?) and not strengthen by literature. All in all, while authors should dampen their conclusions and cite more scientific article to justify their hypothesis and methods, I believe the overall message of the manuscript is of strong interest for Nature Communication. Also, in its current form the manuscript dos not have any discussion - this needs to be added.

Response:

First of all, we appreciate the reviewer’s helpful suggestion. In fact, the original manuscript was submitted to *Nat. Neuroscience* as Brief communication, and later transferred to *Nat. Communications* without changing its format. Therefore, introduction and discussion were

very short due to word limit and citation was limited. While revising this manuscript, we re-wrote the manuscript in full version and included introduction and discussion citing proper references.

In addition, I have a number of comments that I list above, by order of appearance:

1. In most figures the Y axis are often missing, making the figure difficult to read.

Response :

Our apologies for the confusion regarding the graphs. We added Y axis units and marked in the revised manuscript.

2. Statistics: Most of the time, t-tests are not appropriate, author must use multiple comparison tests such as ANOVA, after testing the normal distribution of the data. This is particularly true for behavior data.

Response :

Our apologies for not providing detailed explanation of our statistical analysis. We first used ‘Two-way ANOVA’ or ‘One-way ANOVA’ for all behavioral tests. Then, if there was a significant difference at each period, we performed relevant Two-tails Mann-Whitney test or post hoc. This was specified in the revised manuscript.

3. Bibliography: authors should add much more references to the manuscript in order to strengthen their hypothesis and methods. This might be true for both methods and introductions. Particularly, ChR2 in astrocytes is rarely used – note that the K⁺ control in Fig S9 is welcome and very important, confirming that such opsin might be a good tool to study astrocytes.

Response :

Thank you for this helpful suggestion. We revised the introduction, results and discussion and added relevant references in the revised manuscript.

4. Fig 1: how many animals and cells recorded? Looking at the figure it seems the correlation between hippocampal astrocytes activity and anxiogenic environment exposure is very weak.

Response :

For in vivo imaging, we recorded 87 cells and 129 epochs from 5 mice. We added this information in the revised Methods.

5. Fig 1g: quantification is missing for “center” part of the VR lab., and comparisons should be performed using multiple comparisons tests such as ANOVA if a normal distribution of the data is tested.

Response :

The quantification of “center” part was shown in Fig 1. The reviewer probably intended to point the “closed” part. So, we presented quantification of “closed” part in the revised figure 1 and performed one-way ANOVA as indicated in the revised Supplementary tables.

6. Fig S1: how many cells counted? % ok, n animals ok, but n cells?

Response :

Our apologies for the confusion. We have presented the counted cells in “%”. To be specific, we counted 2174 GFAP⁺ S100 β ⁺ astrocytes in the hippocampus. Of these, 809 astrocyte cells expressed GCaMP6s. Therefore, we noted 36.61% of astrocytes express GCaMP6s (Supplementary fig. 1b).

7. Fig S2: bar graph show that the analysis was performed on only ~30 cells? It is very low according to the statement authors made in the text that 2 populations of astrocytes exist, one pre responsive and one post responsive. How authors exclude that those are not spontaneous random events? This is an example of overstatement.

Response :

We appreciate reviewer's comment. We analyzed more cells in the revised manuscript (Supplementary fig. 2). From a total of 129 cells, 19 cells responded before entering open arm and 78 cells responded after entering open arm. The remaining 32 cells did not respond. The Ca^{2+} trace data showed that peak of Ca^{2+} activity mostly occurs when mice are entering or after entering center area (Figure 1f). Indeed, the number of peaks was increased in VR center compared to corner or closed area (Figure 1g). When we further analyzed the pre-responsive cells in the closed area, the number of astrocyte Ca^{2+} peaks in near-center zone, at which the mice first visually confront anxiogenic open center ahead (Supplementary fig. 3a and b), is much higher than in near-corner zone (Supplementary fig. 3c). Therefore, we do not think astrocytes exhibit calcium activity randomly. These new data were included in Supplementary fig. 3 of the revised manuscript.

8. Fig S3. g and h panels need quantifications. How many EYFP cells were NeuN+? GFAP+? How many GFAP+ were EYFP+? That to assess both efficiency and specificity of the construct. The remarks are also valuable for Fig S1 construct.

Response :

We counted 1206 EYFP cells. Out of it, 1206 cells were GFAP and S100 β -positive. No EYFP-positive cells were NeuN-positive or iba-1-positive. Likewise, we counted 1618 GFAP and

S100 β -positive cells. Out of it, 1206 cells were EYFP-positive. These data are included in the revised manuscript in Figure 2b. In hGFAP-GCaMP6 mice, we counted 2174 GFAP- and S100 β -positive astrocyte cells in hippocampus. Of these, 809 cells expressed GCaMP6s signals. These data are included in Supplementary fig. 1b of the revised manuscript.

9. Fig 2. Because authors used full transgenic mice, the placement of the optic fibers to stimulate astrocytes is critical. Can they provide pictures allowing the reader to assess the right localization of such fibers?

Response :

Thank you for the good suggestion. We indicated the placement of optic fiber with white line in revised Figure 3a. Also, a schematic diagram is provided in Figure 2d. Optic fibers were placed so that the tip of the fibers reaches the top of the stratum orient layer of hippocampus.

10. Fig S4. Most of the y axis labels are missing.

Response :

Once again, our apologies for our mistake. We added Y axis labels in the revised figures.

11. Fig 2d: Accordingly, to the figure and the methods the light stimulation was 5min continuous. This seems to me an enormous stimulation, that should probable be deleterious for the cells, eventually killing some of them. Did control animals also received 5min blue light? Do the authors verified, at least in ex vivo slices, that such stimulation does not kill the astrocytes or neighboring neurons? Methods say the 5min continuous was also applied ex vivo, however panel c shows a ~20s stimulation. Such deleterious effect might explain the long lasting behavioral effects they see, and the absence of “washout” effect (i.e. “after” BL

stimulation) (while the OFT data seem to be a control in themselves).

Response :

A possible tissue-damaging effects of continuous light stimulation was tested by a prior study⁶. In this study, a 15 mW high-power constant light suppressed synaptic activity *in vivo* by increasing tissue temperature, but 3 mW light did not. Therefore, we used the light power from 1 mW ~ 3 mW. In our study, we did not detect any cell death or tissue damage after 5 min of blue light stimulation either in WT or hGFAP-ChR2 mice, *in vivo* as well as *in vitro*. Nor there was any behavioral alteration in WT mice after 5 min light stimulation. To further exclude a possibility of non-specific effects of constant blue light stimulation on synaptic activity, we delivered 5 min constant light again to the prior light-stimulated hippocampal slice. In this experiment, the second light stimulation induced comparable levels of sEPSC frequency increase to that of first light stimulation. This suggest that there was no cell damage after the first 5 min of constant light stimulation. These new data were included in this revised manuscript in Supplementary figure 10g,h.

12. Fig 3. Could authors provide a quantification of the c-fos increase?

Response :

We counted c-Fos-positive cells and added the data in the revised Figure 3b and Supplementary figure 9b.

13. Fig 3: What is the PPADS concentration the authors used? How authors explain the apparent discrepancy between the organotypic slice experiments, showing that ATP is the only gliotransmitter to display an increase in concentration, and the persistence of behavioral effects after PPADS use? One lead might be that PPADS do not block all purinergic receptors.

Interesting hypothesis might be the involvement of other receptors, such as A1 or A2A which can be blocked with CPT or SCH 58261.

Response :

We used 100 μ M PPADS in our experiment that has been indicated in the Method and Figure 4f,g of the revised manuscript. As reviewer pointed out, PPADS completely blocked sEPSC frequency increase in hippocampal slice, yet it partially rescued the behavioral change induced by optogenetic astrocyte stimulation. Microinjection of PPADS *in vivo* inhibited only the duration of stay in open arms in elevated plus maze, but not the time spent in the center area or the speed. This might be due to difference in local concentration of PPADS *in vivo* vs. *in vitro*. The hippocampal area inhibited by PPADS *in vivo* might be restricted compared to hippocampal slice *in vitro*, which might render the difference. As reviewer suggested, other receptors that are less sensitive to PPADS might be involved. In addition, we do not exclude a possibility that other gliotransmitters are involved in the behavioral changes after optogenetic astrocyte stimulation independently of its effects on synaptic change observed. Due to limitation of revision time, also since it is beyond the main scope of our manuscript, we did not completely dissect the reasons, which we hope the reviewer understand. Instead, we fully discussed this issue in this revised manuscript.

14. Line 96 the appeal to fig S3h seems misplaced.

Response :

We apologize for this confusion. This figure shows primary astrocytes expressing ChR2-EYFP. We light stimulated these ChR2-positive astrocytes and confirmed that optogenetic astrocyte stimulation induces intracellular calcium increase (Figure 2c).

Reviewer #3 (Remarks to the Author):

The manuscript entitled “Hippocampal astrocytes modulate anxiety” by Woo-Hyun Cho et al. provides evidence for the involvement of astrocytes in regulating the animal response to the anxiogenic environment. It is showed that upon an increase in intracellular calcium concentration, astrocytes release ATP which in turn increases the sEPSC frequency in neighboring neurons, astrocytes in both the ventral and dorsal hippocampus participate in the modulation of anxiety behaviors. Using a combination of in vivo and in vitro approaches the authors present a cellular mechanism by which astrocyte-neuron communication affects brain activity and behavior. Overall this study is interesting and timely, however, there are several major and minor concerns that authors should address before being considered for publication.

1. The manuscript can be overall improved by better describing and discussing the reported results. Many figure panels are not even cited in the text whereas many others are described all together with one sentence making it hard for the reader to follow the story and to interpret the reported data.

Response :

We revised the original manuscript from a brief report format to an original article format of *Nature Communications*. We fully addressed introduction, results, and discussion, and properly cited references in this revised manuscript.

2. Figure1 and Supp : A GRIN lens implant causes a big amount of brain damage. While I

understand the need for this approach, it would be good to control for reactive astrocytes and consider this technical limitation when describing the results. Moreover, in the method section, there is no mention of the GRIN lens, but rather of a stainless-steel cannula attached to a coverslip. Please comment and adjust.

Response :

We appreciate the reviewer's comment and apologize for the confusion. We mistakenly labeled "GRIN lens" in the figure picture. In fact, we did not implant a GRIN lens to image the CA1 region. We removed the cortical tissue and inserted 2.8 mm-diameter stainless-steel cannula attached to a glass coverslip as noted in the Supplementary information. We used a long working distance (WD = 8 mm) objective, so we did not need to use a GRIN lens.

Moreover, in terms of astrocyte activation followed by cannula implant, as the reviewer pointed, we could discern reactive astrocytes near the surgery site. Therefore, we imaged at 50 ~ 200 μm down from the pyramidal layer, recording stratum radiatum and stratum lacunosum-moleculare layer. In this region, astrocyte morphology or number were comparable with those of control mice. These new data were included as Supplementary Fig. 1f in the revised manuscript.

3. Figure1 and Supp : What is the meaning of measuring the amplitude of calcium peak? This is not a ratiometric calcium indicator and the amplitude can be different for many different reasons that can be independent of the calcium concentration.

Response :

We thank the reviewer for pointing out our mis-interpretation. As reviewer mentioned, $\Delta F/F$ or

amplitude of calcium peak cannot be simply interpreted as $[Ca^{2+}]_i$. Therefore, we changed $[Ca^{2+}]_i$ to ‘intracellular Ca^{2+} activity’ or ‘intracellular calcium event’ throughout the manuscript, and excluded the calcium amplitude data from the revised manuscript.

4. Figure 1 and Supp : It is described that there are no cells active in the closed arms, but from the heat map this does not seem to be the case. From cell-40 to the end there are cells active since the beginning of the peri-event histogram (-3s). Also, the cells grouped as “pre-responsive” to entering the center are active in the middle of the closed arm and represent 50% of the population reported in both conditions (panel I). Please explain why these are not considered as active in the closed arm instead.

Response :

Our apologies for the confusion. We quantified the number of Ca^{2+} peaks per second of all the astrocytes measured while mice were in the center, corner, and closed area, and presented the results in a graph in Figure 1g of the revised manuscript. The number of Ca^{2+} peak was significantly increased in center compared to closed area. As reviewer noted, the ‘pre-responsive’ cells show calcium event in the closed arm. We suspect these cells responded because mice were visually exposed to anxiogenic open arm while nearing the center. During revision, we further divided closed area into two zones, ‘near-corner’ and ‘near-center’, and analyzed astrocyte Ca^{2+} events in these two zones within closed area (Supplementary fig. 3). Even though Ca^{2+} events occurred in closed area, the number of Ca^{2+} peak in “near-center”, where the center is visible, was much higher than in “near-corner”. We think these data support that the hippocampal astrocyte Ca^{2+} activities occur upon exposure to anxiogenic environment. These new data are included in this revised manuscript as Supplementary fig. 3.

5. Figure1 and Supp : The tracking ball data (i.e., distance traveled, speed) and the reward release time should be reported to make sure the astrocytes are not active in correlation to other variables.

Response :

We appreciate the reviewer’s suggestion. We further analyzed the correlation between the distance traveled and the astrocyte Ca^{2+} activity, and did not find any correlation. These data support that astrocyte are not active in correlation to distance traveled or speed. We presented these data in this letter for reference (Reference Figure 2). We allowed water reward only during training session, and recorded astrocyte calcium activity without any reward during test session.

Reference figure 2.

Representative Ca^{2+} trace during exploration in VR condition. Black trace indicates Ca^{2+} trace, Green trace indicates moving distance, Yellow bar showed active movement.

6. Figure1 and Supp : The analysis of the calcium activity has been limited to the cell soma, what is the degree of calcium activity in the main branches and microdomains? Is there an increase in relation to the mouse position in the VR environment, distance traveled, speed or reward?

Response :

We analyzed Ca^{2+} responses in astrocyte branches and soma. Although we could not exactly separate main branches vs. microdomains, we found that Ca^{2+} responses also occur in astrocyte branches in anxiogenic environment (Reference figure 3). However, the calcium signal traces in branches do not completely match with that of cell soma. Therefore, the correlation and significance of calcium signals in soma vs. branches in anxiolytic behavior need to be characterized, which we left for future study. These data are presented as Reference figure below.

Reference figure 3.

Representative imaging of Ca^{2+} response and trace of Ca^{2+} activity.

7. Figure1 and Supp : At line 74 Supp figure 1 is not described.

Response :

Our apologies for missing the detailed explanation. We have provided a detailed description in the Results section (page 4) of the revised manuscript.

8. Figure1 and Supp : The legend of Fig Supp figure 1 seems to be wrong for panel a. The ChR2 expressing astrocytes are described, but they are GCaMP6s expressing astrocytes.

Response :

Thank you for noting our mistake. We corrected the legend accordingly in the revised manuscript.

9. Figure1 and Supp : The whole analysis in Supp figure 2 is not clear and not described in the main text. From the legend, it seems like different time windows were used to define cells compared to the one reported in Fig1. Please explain.

Response :

Our apologies for missing detailed explanation. We provided a detailed explanation in the Results section of the revised manuscript (page 5). Also, to obviate confusion, we re-analyzed the Figure 1 data and presented under the same time windows. As the time window was changed from 6 seconds to 20 seconds, the cell proportion changed slightly. Still, the data confirmed that the hippocampal astrocytes showed a calcium increase when mice entered the center area. These updated data are presented in Figure 1h-i in the revised manuscript.

10. Figure 2 and Supp : What is the effect of ChR2 stimulation on the calcium activity of astrocytes in vivo? Does the optogenetic-induced calcium activity last longer than the

behaviorally-elicited activity?

Response :

Astrocyte Ca^{2+} elevation during blue light delivery was returned to basal level within a couple of seconds when the light stimulation was turned off *in vitro* (Figure 2c). Meanwhile, in the VR, Ca^{2+} activity peaked in 1 second, but took ~ 10 seconds to return to the basal level. Therefore, it seems that behaviorally elicited astrocyte calcium activities last longer than the optogenetically-elicited calcium activity. However, since we could not measure optogenetically-elicited astrocyte Ca^{2+} activities *in vivo* due to technical limitation, we could not directly compare the behaviorally- vs. light-induced astrocyte calcium duration *in vivo*.

11. Figure 2 and Supp : Although reported as an important effect throughout the paper, the long-lasting behavioral change in the second 5min after the optogenetic stimulation is not further discussed. What is the possible reason?

Response :

As commented by reviewer, a key feature of optogenetic astrocyte activation is that it exerts long-lasting behavioral effects. The increase in the time spent in open arms and center, speed, and the exploratory drive in OFT were all maintained after the light was turned off (Figure 2). In addition, PPADS blocked the long-lasting behavioral effects in time in the center area or speed in the EPM (revised Figure 4k,l). There is no clear explanation for these long-lasting anxiolytic effects. Considering the pivotal role of astrocytes in synaptic plasticity, it is conceivable that astrocyte-derived ATP and its downstream signaling may render long-lasting behavioral effects by influencing DG synaptic plasticity of mossy fiber-CA3 synapses, which warrants future investigation. This was discussed in the revised manuscript.

12. Figure 2 and Supp : Since panels 2h-i describe an increased mean speed upon optogenetic stimulation of astrocytes, I even strongly recommend analyzing the mean speed in the VR experiments and see if there is a correlation between calcium activity and traveling speed.

Response :

As shown in the Reference Figure 2 provided above, there was no correlation between the distance traveled and the Ca^{2+} response. The distance traveled variable is the same as the traveling speed because it is the distance traveled per second.

13. Figure 2 and Supp : Do the neurons in both the dorsal and ventral parts of the hippocampus respond in the same way to the astrocytes manipulation?

Response :

According to our data, neurons in both the dorsal and ventral parts of the hippocampus responded similarly to the optogenetic astrocyte stimulation. Astrocyte activation in dorsal and ventral hippocampus induced c-Fos in both dorsal and ventral neurons (Supplementary fig. 9) and led to comparable behavioral outcome (Supplementary fig. 6,7,8). Also, the electrophysiological impact on the dorsal and ventral hippocampal neurons was the same: optogenetic astrocyte activation increased sEPSC frequency in both dorsal and ventral area (Supplementary fig. 10a-d). In this regard, we expect that astrocytes affect dorsal and ventral neurons with similar mechanism. We included these new data in revised Supplementary fig. 9 and 10.

14. Figure 2 and Supp : How do you explain the results in Supp figure 5? What is the percentage of time spent in the center versus the sides of the open field arena?

Response :

We analyzed the percentage of time spent in the center in the OFT, and presented here as Reference Figure 4. In these data, optogenetic astrocyte stimulation did not increase the percentage of time spent in the center. Still, it significantly increased the exploratory drive in open field test (Figure 2 and Supplementary fig. 7).

Reference Figure 4. Percentage of time in center

Two-way ANOVA, $F(1,27) = 1.208$, $p = 0.281$

15. Figure 2 and Supp : It is not clear to me how the elevated calcium activity reported in figure 1 in response to an anxiogenic environment correlates to the anxiolytic effect of elicited calcium activity reported in figure 2. Please clarify.

Response :

In this study, we did not obtain clear evidence demonstrating the causality of hippocampal astrocyte calcium activation in anxiogenic environment and anxiolytic behaviors. Although the physiological relevance or the outcome of the hippocampal astrocyte calcium signals in anxiogenic environment is not completely elucidated, we revealed that astrocyte calcium activation accompanied ATP release, which in turn increased synaptic activity of DG granule cells. It was previously reported that DG granule cell activation exerts anxiolytic effect⁷.

Therefore, we suspect that hippocampal astrocyte activation in anxiogenic environment may contribute to overcome or reduce innate anxiety by affecting hippocampal synaptic activity. We addressed this issue in the Discussion section of the revised manuscript.

16. Figure 2 and Supp : From the example reported in panel j, it does not seem like the hGFAP-ChR2 mice are spending more time in the center compared to controls.

Response :

In open field test, although optogenetic astrocyte activation did not induce increased time spent in the center (Reference Figure 4), it increased total distance moved and speed (Figure 2 and Supplementary figure 7), thus increased exploratory drive in open field test.

17. Figure 2 and Supp : In Supp figure 6 please compare the % of time spent in the light chamber also in the session before and after the light stimulation. The n should also increase as in Supp figure 6d, it seems that 3/5 mice are behaving in opposite ways (CTRL spending more time in the stim side and hGFAP-ChR2 spending less time in the stim chamber) and the remaining 2 mice do the opposite. Moreover, no statistical tests are reported in the legend.

Response :

We have performed additional experiments, and increased mice number for real-time place preference to 10 mice per group. We presented these updated data as Supplementary fig. 8 in the revised manuscript. Also, we specified the statistical method in the Supplementary figure legend and Supplementary table.

18. Figure 3 and Supp : How does the time scale of the sEPSCs increased frequency fit the behavioral effects lasting for an extra 5min after the astrocyte stimulation?

Response :

We agree with the reviewer's comment. The increase of sEPSC frequency did not last 5 min after the optogenetic astrocyte activation, but it gradually decreased during light stimulation and returned to basal level. Therefore, this result does not explain the maintenance of the behavioral consequences. We admit that we don't have clear mechanistic explanation for the long-term behavioral impact of astrocyte stimulation. Still, our data show PPADS could block the synaptic change and also some of the long-term behavioral effects. These data indicate that activated astrocyte-derived ATP release and its downstream purinergic receptor activation on hippocampal neurons mediate the long-term behavioral effect. We can speculate the brief sEPSC frequency increase in DG granule cells during astrocyte activation somehow causes the long-term behavioral effects by affecting synaptic plasticity within hippocampal circuits. Alternatively, we do not exclude the possibility that purinergic receptor activation affects hippocampal synaptic activity or remodeling independently of sEPSC frequency increase, which eventually reads to the long-term anxiolytic effects, which warrants future investigation.

19. Figure 3 and Supp : The experiments reported in Supp fig.9 are based on a study in which is described the increase in extracellular K⁺ level upon astrocytes optogenetic excitation in the striatum. However, the maximum reported concentration of K⁺ (7.5 mM) is based on 1000 light pulses of 25ms, which is 25s light stimulation, without considering an inter-pulse-interval. In the present manuscript, the stimulation time is 5 min with continuous light. To compare the data reported here with the previous study there should be first an understanding of what is the maximum concentration of extracellular K⁺ after 5 min stimulation and then study the effect of that concentration on frequency and amplitude of sEPSC.

Response :

We appreciate the reviewer's helpful comment. In our study, we could not exactly measure the *in vivo* extracellular concentration of K^+ upon optogenetic astrocyte stimulation due to technical difficulty. Instead, we increased the extracellular K^+ concentration in ACSF up to 9 mM to test if higher concentration of K^+ recapitulate the electrophysiological changes after optogenetic astrocyte stimulation. Our data show that both amplitude and frequency of sEPSC were elevated in 9 mM K^+ ACSF (Supplementary fig. 11 k and l), while these alterations were not observed in 7 mM of K^+ (Supplementary fig. 11 g and h). Upon optogenetic astrocyte stimulation, only frequency but not amplitude of sEPSC increased (Figure 3e, f). Therefore, increase of extracellular K^+ , even at higher concentration, fail to recapitulate the synaptic activity changes provoked by optogenetic astrocyte stimulation. Furthermore, such synaptic activity change was completely abrogated by PPADS treatment. In this regard, we argue that the synaptic as well as the behavioral effects are due to astrocyte-derived ATP rather than K^+ increase. We included these new data in Supplementary fig. 11 of the revised manuscript.

20. Figure 3 and Supp : Does the neuronal firing rate increase with astrocyte optogenetic stimulation alone?

Response :

To determine whether optogenetic astrocyte stimulation increases excitability of hippocampal neurons, we applied slow ramps to verify the amount of current needed to induce action potentials. Although the threshold to trigger action potential was reduced in hGFAP-ChR2 mice, the excitability in the last second was not different between control and hGFAP-ChR2 mice, which show the neuronal firing rate did not increase with optogenetic astrocyte activation. These new data were included in the revised manuscript as Supplementary fig. 11a-c.

21. Figure 3 and Supp : The light stimulation for ATP, Glutamate, and D/DL-serine measurement is twice the one used in vivo. Please explain why.

Response :

For measurement of gliotransmitter release *in vitro*, we followed the stimulation protocol published previously⁸. In that study, they light-stimulated astrocytes for 20 minute and performed ATP and mRNA assay. Since *in vivo* and *in vitro* conditions are different, we conducted the experiment following the literature. We have also stimulated astrocytes for 5 min, and found ATP is also released in this condition.

22. Figure 3 and Supp : In panel f the cumulative probability looks significantly different even though the mean is not. Please test and report.

Response :

We have performed additional electrophysiological experiments to add more cells, and performed the Kolmogorov-Smirnov (KS) test to assess statistical significance of cumulative plots. We revised the figures (Figure 4f), and added statistic information in figure legend and supplementary tables.

23. Figure 3 and Supp : Also in this case, in Supp figure 10 there is a major point about the increased speed and total travel distance which is not blocked by PPADS injection.

Response :

In our data, PPADS *in vivo* inhibited only the duration of stay in open arms in elevated plus maze. The time spent in the center area or the speed were not affected by PPADS. Also, in the open field test, there were no differences between PPADS-injected and control groups.

Although we identified significant ATP release by astrocyte activation, we do not exclude the possibility of other gliotransmitters involvement. We can speculate that ATP release and subsequent purinergic receptor signaling in neurons may render mice to face potential threats, but the increase in speed or exploratory drive may require other yet unknown pathways which may or may not be dependent of ATP. We have addressed this issue in the Discussion section of the revised manuscript.

25. All cumulative probability distribution should be analyzed using the Kolmogorov-Smirnov test. The mean frequency analysis is not sufficient as the distributions in many cases look different even though the mean is not.

Response :

We appreciate the reviewer's comment. As the reviewer's previous comment (comment #22), we performed KS test to cumulative probabilities in all sEPSC data. We presented in the revised Figures and added statistic information in figure legend and supplementary tables.

26. It is not always clear which statistical test has been performed on the data. For some analysis (i.e., Figure 2 and in Figure 3) it is necessary to know what test has been used to define statistical significance. Due to the presence of many graphs, it becomes difficult to know for sure by reading the legend and the statistical methods.

Response :

We specified the statistical test used in each figure legends of the revised manuscript. Also we made a supplementary statistic table that contains this information.

27. The t-value for all the t-test and the F value for all the ANOVA need to be reported.

Response :

We included this information in figure legends as well as in the supplementary statistic table.

References

1. Komada, M., Takao, K. & Miyakawa, T. Elevated plus maze for mice. *J Vis Exp* (2008).
2. Perea, G., Yang, A., Boyden, E.S. & Sur, M. Optogenetic astrocyte activation modulates response selectivity of visual cortex neurons in vivo. *Nat Commun* **5**, 3262 (2014).
3. Gourine, A.V., *et al.* Astrocytes control breathing through pH-dependent release of ATP. *Science* **329**, 571-575 (2010).
4. Takata, N., *et al.* Optogenetic astrocyte activation evokes BOLD fMRI response with oxygen consumption without neuronal activity modulation. *Glia* **66**, 2013-2023 (2018).
5. Shen, W., Nikolic, L., Meunier, C., Pfrieger, F. & Audinat, E. An autocrine purinergic signaling controls astrocyte-induced neuronal excitation. *Sci Rep* **7**, 11280 (2017).
6. Owen, S.F., Liu, M.H. & Kreitzer, A.C. Thermal constraints on in vivo optogenetic manipulations. *Nat Neurosci* **22**, 1061-1065 (2019).
7. Kheirbek, M.A., *et al.* Differential control of learning and anxiety along the dorsoventral axis of the dentate gyrus. *Neuron* **77**, 955-968 (2013).
8. Nam, Y., *et al.* Reversible Induction of Pain Hypersensitivity following Optogenetic Stimulation of Spinal Astrocytes. *Cell Rep* **17**, 3049-3061 (2016).

REVIEWER COMMENTS

Reviewer #1 (Remarks to the Author):

In the revised manuscript, Cho et al. included additional data and discussion to address my comments. Overall, the revision is satisfactory. However, there are still several key points that the authors should consider before their work being published at Nature Communications.

Major points:

1. To address my concern of using ChR2 to activate astrocytes, which are not electrically excitable cells and therefore cannot generate action potentials upon ChR2 activation, the authors clarified that their definition of "astrocytic activation" is based on "intracellular astrocyte calcium activation". However, there is no evidence provided to demonstrate that astrocytic ChR2 activation indeed increased astrocyte intracellular calcium. This data will be necessary to support their claim. It is also important to analyze the kinetics of astrocyte calcium changes, if any, because this may provide insights to the puzzling observations that behavioral changes (e.g. increased exploration in the open arm of EPM and in the OFT) lasted long after the light was off in mice expressing ChR2 in astrocytes.
2. In Figure 2, mice expressing ChR2 were subjected to EPM and OFT to evaluate the anxiety-like behavior. However, time spent in the center of the OFT, which is a standard measure of anxiety levels, was missing from the analysis.
3. Figure 4 and the text description are completely mismatched: there is no explanation of Figure 4c-d in the text, and all the panel call-outs are wrong. The interpretation of results is difficult to the readers.
4. In Figure 4, the authors used an ATP Bioluminescence Assay Kit to measure the ATP release from the hippocampal brain slices. However, this assay does not provide cell-type specificity, i.e. chemicals were collected from all cell types, not just astrocytes. The authors should consider to use genetic ATP sensors targeted to astrocytes to address this concern (Lobas MA. et al. 2019, Nature Communications).

Reviewer #2 (Remarks to the Author):

The authors have addressed all of my remarks and comments. I appreciate the improved bibliography and discussion. However, I highly recommend author to work one more time on those specific aspects: indeed, to show that hippocampal astrocytes play an active role in modulating neuronal networks and related behaviors in this structure is of tremendous importance. Authors must not underestimate the potential impact of such result in the field (but still not overinterpret the results!).

I have here three points that might help the authors to expand the scope and impact of their manuscript:

- If astrocytes play an active role in anxiety, one can guess they will do the same for other hippocampal-related behaviors, from social interaction to working memory.
- In addition, I would like to see what authors think about ATP-mediated gliotransmission: what are there hypothesis regarding this vs glutamate / D-serine gliotransmission in other structures? Why should we have such striking differences? Can they imagine a role for different gliotransmission in hippocampus?
- Authors slightly touch my last point: endogenous activators of hippocampal astrocytes. As they explained, oxytocin was shown to modulate amygdala astrocytes, and very recently, J. Stern

group published a manuscript showing oxytocin receptor expression on hippocampal astrocytes. Can they foresee a discussion link? What might be the other endogenous modulators that scientific colleagues might be interested in, from dopamine to other neuropeptides?

In addition, I have a minor concern with the current presentation of the results: throughout most of the manuscript, authors say "hippocampus" without specification of the region. I guess they might indicate more clearly which region was targeted – dentate gyrus, if I am right.

Finally, I did not have time to go through the detailed material and methods. I apologize for this and recommend the authors to carefully review this important part of the manuscript.

Reviewer #3 (Remarks to the Author):

After reading the rebuttal to all the reviewers' comments, I believe a great deal of supporting experiments and detailed analysis have been added to clarify all raised concerns. I believe this work will generate many useful questions to investigate in the future regarding the role of astrocytes in controlling animal behaviors.

In my opinion the paper is ready for publication and there is no need for any further experiments.

Good luck

Reviewer #1 (Remarks to the Author):

In the revised manuscript, Cho et al. included additional data and discussion to address my comments. Overall, the revision is satisfactory. However, there are still several key points that the authors should consider before their work being published at Nature Communications.

Major points:

1. To address my concern of using ChR2 to activate astrocytes, which are not electrically excitable cells and therefore cannot generate action potentials upon ChR2 activation, the authors clarified that their definition of “astrocytic activation” is based on “intracellular astrocyte calcium activation”. However, there is no evidence provided to demonstrate that astrocytic ChR2 activation indeed increased astrocyte intracellular calcium. This data will be necessary to support their claim. It is also important to analyze the kinetics of astrocyte calcium changes, if any, because this may provide insights to the puzzling observations that behavioral changes (e.g. increased exploration in the open arm of EPM and in the OFT) lasted long after the light was off in mice expressing ChR2 in astrocytes.

Response:

In our original manuscript, we provided data showing intracellular Ca^{2+} activation in primary cultured astrocytes upon ChR2 activation. To further confirm this phenomenon in adult hippocampal astrocytes, we injected GFAP-jRGECO1a, an astrocyte-specific red-shifted Ca^{2+} indicator, into the hippocampus of adult hGFAP-ChR2 mice. Next, we measured the intracellular astrocyte calcium signal in acute hippocampal slices while delivering blue light stimulation (Fig 2d, e). We observed significantly increased jRGECO1a fluorescence during optogenetic astrocyte

stimulation, demonstrating that ChR2 stimulation indeed triggers intracellular Ca²⁺ activation in adult hippocampal astrocytes (Fig. 2f). In this experiment, the ChR2-activated astrocyte calcium signal returned to the basal level upon light removal. This result indicates that the astrocyte calcium signal kinetics cannot explain the long-lasting behavioral effects of optogenetic hippocampal astrocyte activation. We included these data in Figure 2d-f in the revised manuscript.

2. In Figure 2, mice expressing ChR2 were subjected to EPM and OFT to evaluate the anxiety-like behavior. However, time spent in the center of the OFT, which is a standard measure of anxiety levels, was missing from the analysis.

Response:

When we measured the time spent in the center of the OFT, optogenetic astrocyte stimulation did not significantly increase the percentage of the time spent in the center. We have already provided this result to the reviewer during our 1st revision as reference Figure 4 in the response letter.

Mouse anxiety status can be measured using diverse behavioral tests, including EPM and OFT, and mice behavior could vary depending on the anxiogenic stimuli or physical/emotional condition of the tested mice. For instance, a previous study reported that thirsty rats exhibited significant anxiety behavior in EPM but not in OFT (Rebolledo-Solleiro *et al.*, *Physiology & Behavior*, 2013). In our data, optogenetic hippocampal astrocyte stimulation significantly increased the time in the center and open arm in EPM (Figure 2). It also significantly increased the exploratory drive in the open field test (Figure 2) and the distance traveled in the center area (Supplementary figure 7b). Although, somehow, no significant difference in the time spent in the center of the OFT was observed, our behavioral data show that optogenetic hippocampal astrocyte activation confers

anxiolytic behaviors overall.

3. Figure 4 and the text description are completely mismatched: there is no explanation of Figure 4c-d in the text, and all the panel call-outs are wrong. The interpretation of results is difficult to the readers.

Response:

We believe that the reviewer probably confused the current and previous versions of the figure and manuscript because we have provided a clear description for Figure 4c and d in lines 277–280, page 13, in the previous version of the revised manuscript. We have ensured the other figures and text description match.

4. In Figure 4, the authors used an ATP Bioluminescence Assay Kit to measure the ATP release from the hippocampal brain slices. However, this assay does not provide cell-type specificity, i.e. chemicals were collected from all cell types, not just astrocytes. The authors should consider to use genetic ATP sensors targeted to astrocytes to address this concern (Lobas MA. et al. 2019, Nature Communications).

Response:

We appreciate the reviewer's suggestion. The reviewer suggested using astrocyte-specific iATPSnFRs (genetic ATP sensor) to define the cellular ATP source. However, iATPSnFRs increase their fluorescence upon binding extracellular ATP. Therefore, even if astrocyte-specific iATPSnFRs were activated, it does not necessarily prove that they are the cellular ATP source.

Therefore, we believe that this additional experiment does not have a critical merit.

Reviewer #2 (Remarks to the Author):

The authors have addressed all of my remarks and comments. I appreciate the improved bibliography and discussion. However, I highly recommend author to work one more time on those specific aspects: indeed, to show that hippocampal astrocytes play an active role in modulating neuronal networks and related behaviors in this structure is of tremendous importance. Authors must not underestimate the potential impact of such result in the field (but still not overinterpret the results!).

I have here three points that might help the authors to expand the scope and impact of their manuscript:

1. If astrocytes play an active role in anxiety, one can guess they will do the same for other hippocampal-related behaviors, from social interaction to working memory.

Response:

We appreciate the reviewer's comment. Indeed, recent publications have indicated the critical role of hippocampal astrocytes in memory functions (Adamsky *et al.*, *Cell*, 2018 & Kol *et al.*, *Nat Neurosci*, 2020). In our study, although optogenetic hippocampal astrocyte activation did not impair T-maze behavior (Supplementary fig.8 a-c), it may have affected memory performance accompanying anxiety. Moreover, hippocampal neurons project to distal brain regions, such as the prefrontal cortex, and regulate mouse social behaviors (Sun *et al.*, *iScience*, 2020).

Considering that optogenetic astrocyte stimulation enhanced excitatory synaptic transmission in our study, it is possible that hippocampal astrocytes selectively modulate neuronal outputs to the prefrontal cortex during social behavior and thereby affect mouse social behaviors, which warrants future investigation. We have addressed this possibility in the Discussion section of the revised manuscript.

2. In addition, I would like to see what authors think about ATP-mediated gliotransmission: what are there hypothesis regarding this vs glutamate / D-serine gliotransmission in other structures? Why should we have such striking differences? Can they imagine a role for different gliotransmission in hippocampus?

Response:

We thank the reviewer for this comment. As the reviewer pointed out, activated astrocytes release various gliotransmitters including glutamate (Mahmoud *et al.*, *Cells-Basel*, 2019) and D-serine (Koh *et al.*, *Biol Psychiatry*, 2022), which are critical for the proper regulation of synaptic transmissions and behaviors. The hippocampus is involved not only in regulating emotions but also various memory-related behaviors. Astrocyte-derived glutamate and D-serine are already known to regulate hippocampus-dependent memory performance by affecting neuronal long-term potentiation (Cheung *et al.*, *Nat Communi*, 2022 & Koh *et al.*, *Biol Psychiatry*, 2022). Although we did not detect a significant glutamate or D-serine increase in our anxiogenic mouse behavioral paradigm, we cannot exclude the possibility that other gliotransmitters are released in response to other stimuli or other behavioral contexts and affect other hippocampus-dependent behaviors. We have addressed this aspect in the Discussion section of the revised manuscript (pages 18 and 19).

3. Authors slightly touch my last point: endogenous activators of hippocampal astrocytes. As they explained, oxytocin was shown to modulate amygdala astrocytes, and very recently, J. Stern group published a manuscript showing oxytocin receptor expression on hippocampal astrocytes. Can they foresee a discussion link? What might be the other endogenous modulators that scientific colleagues might be interested in, from dopamine to other neuropeptides?

Response:

We would like to thank the reviewer for this suggestion. Astrocytes express receptors for diverse neuromodulators, including oxytocin, dopamine, and serotonin. Considering that hippocampal astrocytes express oxytocin receptors (Althammer *et al.*, *J Neuroendocrinol*, 2022), as the reviewer mentioned, it is conceivable that oxytocin may exert its anxiolytic effects by activating hippocampal astrocytes. Furthermore, astrocyte dopamine and serotonin receptors regulate synaptic transmission and behaviors. For instance, prefrontal astrocytes control dopaminergic homeostasis that leads to cognitive impairments (Petrelli *et al.*, *Mol Psychiatry*, 2020), and astrocyte dopamine D1 receptors in the nucleus accumbens enhance amphetamine-induced mouse locomotion (Corkrum *et al.*, *Neuron*, 2020). Moreover, hippocampal astrocytes express serotonin receptor 4 that regulates astrocyte morphological changes and synaptic transmissions (Fang *et al.*, *J Neuroinflamm*, 2022). From this perspective, our data suggest a novel possibility that dopamine and serotonin, well-known neuromodulators regulating mood and emotion, may exert their effects by regulating hippocampal astrocyte activity, in addition to their direct effects on neuronal dopamine/serotonin receptors. We have discussed this possibility in the revised manuscript (page 17).

In addition, I have a minor concern with the current presentation of the results: throughout most of the manuscript, authors say “hippocampus” without specification of the region. I guess they might indicate more clearly which region was targeted – dentate gyrus, if I am right.

Response:

We appreciate this point. We measured the astrocyte calcium activity in the stratum radiatum and stratum lacunosum-moleculare layers in the hippocampus. The optic fiber was implanted above either the dorsal or ventral CA1 to target the stratum radiatum and stratum lacunosum-moleculare layers in the hippocampus. We have included this information in the revised manuscript on page 6, line 19 and page 9, lines 12 and 13.

REVIEWERS' COMMENTS

Reviewer #1 (Remarks to the Author):

The authors have addressed my questions and concerns.

Reviewer #2 (Remarks to the Author):

The authors addressed all my points. I am convinced this article will be of high impact and recommend its publication in nature communication.

REVIEWERS' COMMENTS

Reviewer #1 (Remarks to the Author):

The authors have addressed my questions and concerns.

- We appreciate reviewer's positive evaluation of our manuscript.

Reviewer #2 (Remarks to the Author):

The authors addressed all my points. I am convinced this article will be of high impact and recommend its publication in nature communication.

- We appreciate reviewer's positive evaluation of our manuscript.